*Method*

# CANTAO: guiding clustering and annotation in single-cell RNA sequencing using average overlap

Christopher Thai[1,2], Amartya Singh [ID][1,2], Daniel Herranz [ID][1,3,4 ✉] & Hossein Khiabanian [ID][1,2,5,6 ✉]

## Abstract

**Single-cell RNA sequencing allows defining cellular identities based on transcriptional similarity using unsupervised clustering. However, a single clustering resolution may not yield groups of cells that represent both broad, well-defined populations and smaller subpopulations simultaneously. Therefore, when cell identities are not known prior to sequencing, robust comparison and annotation of inferred de novo clusters remains a challenge. Here, we introduce CANTAO, in which we propose the average overlap metric to define the distance between single-cell clusters by comparing ranked lists of differentially expressed genes in a top-weighted manner. We benchmark CANTAO in truth-known datasets comprised of similar yet distinct cell populations and show that evaluating clusters with average overlap results in a consistent, precise, and biologically meaningful recapitulation of true cell identities. We then analyze unsorted mouse thymocytes and characterize stages of T-cell development in the thymus, including minor populations of double-negative (CD4-CD8-) T cells that are difficult to confidently detect among unsorted single cells. We demonstrate that CANTAO enables robust, reproducible characterization of single-cell data and clarifies biological interpretation of underlying identities in homogeneous populations.**

**Keywords** Clustering; scRNA-seq; Cell Annotation; Thymocytes; T-cell Development
**Subject Category** Computational Biology

## Introduction

Understanding complex biological processes, such as hematopoiesis or tumorigenesis, requires an accurate identification of the types of cells present in a tissue sample and a mapping of their interactions with one another. Bulk RNA sequencing methods measure average gene expression across all cells (Li and Wang, 2021) and cannot measure cell-to-cell gene expression variation that may arise due to functional differentiation, such as those during T-cell development, or time-dependent processes that occur across tumor clonal evolution. Meanwhile, single-cell RNA sequencing (scRNA-seq) technologies have been developed to address these challenges, leading to an improved understanding of molecular processes in complex diseases (Luecken and Theis, 2019).

Defining cell types using unsupervised clustering algorithms based on transcriptional similarity is a powerful application of scRNA-seq. There are several steps involved in the computational analysis of scRNA-seq data (Haque et al, 2017; Hwang et al, 2018; Luecken and Theis, 2019), including initial quality control, normalization, clustering, and identifying differentially expressed genes, which are considered as marker genes (Pullin and McCarthy, 2024) for each inferred cluster. One can then analyze these marker genes and consult the literature and reference databases to assign biological identities to each cluster. These steps are currently implemented as a complete workflow in commonly used analytical packages like Seurat (Hao et al, 2024; Satija et al, 2015) and Scanpy (Wolf et al, 2018).

Still, challenges remain in clustering of scRNA-seq data stemming from the nature of measurements from single cells (Kiselev et al, 2019). Cell identities are not known before sequencing without a prior sorting process, and noise in measuring gene expression, especially for those expressed in small subpopulations, can confound cell clusters inferred by unsupervised community detection algorithms. In addition, parameters that determine the number and membership of clusters, such as the resolution in Leiden or Louvain methods (Blondel et al, 2008; Traag et al, 2019), are applied globally across an entire dataset, despite the possible presence of distinct subpopulations with varying sizes. To avert over- or under-clustering and subsequent mischaracterization of novel cell populations and their marker genes, one can directly annotate cells using their expression of pre-defined sets of marker genes (Aran et al, 2019; Hou et al, 2019; Ianevski et al, 2022; Pasquini et al, 2021). However, direct annotation is constrained to identities present in reference datasets and may miss novel biological populations uncharacterized in the literature. Meanwhile, solutions for optimizing clustering parameters have been proposed (Duo et al, 2018; Kim et al, 2019; Patterson-Cross et al, 2021; Peyvandipour et al, 2020; Yu et al, 2022), but a consistent biological interpretation of inferred clusters remains an unresolved necessary step.

Inferring similar cell state or identity for clusters that share key marker genes is analogous to a subjective call on two lists' similarity

[1]Rutgers Cancer Institute, Rutgers University, New Brunswick, NJ 08901, USA. [2]Center for Systems and Computational Biology, Rutgers Cancer Institute, Rutgers University, New Brunswick, NJ 08901, USA. [3]Department of Pharmacology, Rutgers Robert Wood Johnson Medical School, Rutgers University, Piscataway, NJ 08854, USA. [4]Department of Pediatrics, Rutgers Robert Wood Johnson Medical School, Rutgers University, New Brunswick, NJ 08901, USA. [5]Department of Pathology and Laboratory Medicine, Rutgers Robert Wood Johnson Medical School, Rutgers University, New Brunswick, NJ 08901, USA. [6]Present address: Regeneron Genetics Center, Regeneron Pharmaceuticals, Tarrytown, NY 10591, USA. ✉E-mail: dh710@cinj.rutgers.edu; h.khiabanian@rutgers.edu

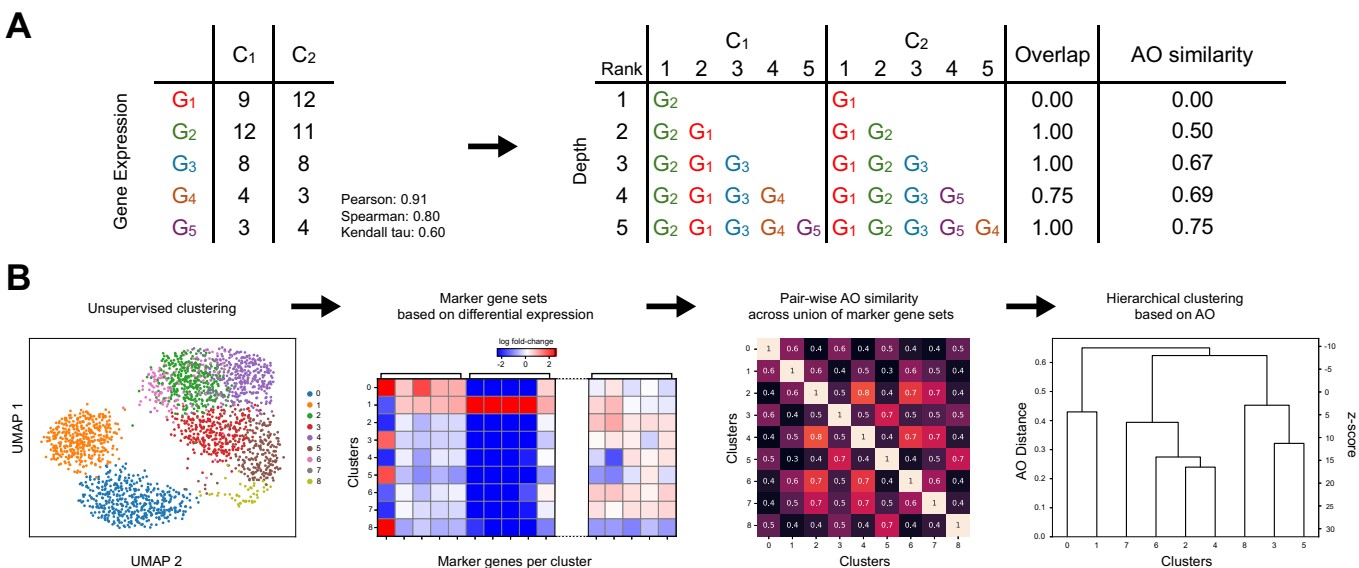

**Figure 1. CANTAO workflow for quantitative comparison of inferred single-cell clusters in scRNA-seq data.**

(A) An example of calculating the average overlap distance between two ranked lists and comparing with correlation-based metrics. Differentially expressed genes (G1 to G5) are ranked in clusters C1 and C2, and the overlap between the ranked lists is calculated at each depth. The average overlap of the entire list is the mean of all overlaps. The Pearson, Spearman, and Kendall tau correlation coefficients are shown for comparison. (B) A schematic demonstrating the workflow for hierarchical clustering using average overlap, as proposed in CANTAO, in computational scRNA-seq analysis. Following community detection and marker gene identification, pair-wise AO similarities are computed for each cluster pair. These AO scores are the metrics used to perform hierarchical clustering of the inferred clusters, and the resulting cluster tree allows for assessing their similarity at a glance.

based on the presence of a few key members. Conversely, the mere presence of the same genes in two or more clusters' marker gene sets may not by itself indicate the same cell state or identity. Differences in the magnitude of differential expression in each cluster, and by extension, differences in the rankings of marker genes, may indicate a relevant differentiation trajectory or novel subpopulation. Yet, methods are lacking for the quantification of marker gene set similarity between single-cell clusters to accurately inform the annotation of cell identity.

Metrics that compare two lists based on the ranking of their elements address this challenge by providing a measure to compare single-cell marker gene sets from cluster-based differential expression analyses. While rank-based methods for scRNA-seq analyses have been developed (Oulas et al, 2024; Vargo and Gilbert, 2020; Xu et al, 2021), none specifically address cluster similarity based on differential expression, especially when clusters are computationally determined in an unsupervised manner through community detection algorithms like Leiden or Louvain. To date, no quantification of the similarity of differentially expressed genes and their rankings has been employed in downstream scRNA-seq analysis. Here, we introduce CANTAO (**C**lustering and **AN**notation using **T**ranscriptomic **A**verage **O**verlap), in which we provide a definition for similarity between single-cell clusters based on a rank-based metric called average overlap (Webber et al, 2010) and show that it can be used to calculate the significance of related cell identities in a truth-known dataset with consistency, precision, and meaningful biological interpretation. We then demonstrate the utility of CANTAO for guiding the clustering and annotation of cells in the thymus, revealing individual stages of thymocyte development.

## Results

### Average overlap metric for cluster marker gene comparison

To compare single-cell clusters derived from unsupervised community detection algorithms, we first define each cluster's marker gene set by its most differentially expressed genes relative to the rest of the cells. We then rank genes in each set based on the significance of differential expression to obtain cluster marker genes and hypothesize that clusters that share marker genes with similar rankings are more likely to have similar cell identity and/or state.

To test this hypothesis, we propose using CANTAO, designed based on a rank-based metric called average overlap (AO) (Webber et al, 2010), which is a top-weighted measure of ranked list similarity. In the context of marker gene sets, AO weighs differences in the rankings of the most differentially expressed genes higher than for the genes lower in the set. AO is defined by the mean of the overlaps between two ranked lists calculated at a range of depths into the lists (Fig. 1A). AO distances range from 0 to 1, from completely dissimilar to completely identical lists. When calculated over randomly shuffled lists, AO closely follows a normal distribution (Appendix Fig. S1) centered around the midpoint AO distance of 0.5. This distribution provides a statistical means for assigning significance to AO distances by defining AO z-scores and calculating a likelihood for rank similarity of two marker gene sets. Using these z-statistics, randomly shuffled marker sets containing the same genes will have an AO z-score of 0 on average, whereas similarly ranked sets will have increasingly positive AO z-scores. In contrast, and as marker gene sets become increasingly

dissimilar, the AO z-scores become smaller and eventually become increasingly negative.

Our proposed workflow in CANTAO involves the use of AO as the underlying distance used to compute a hierarchical clustering of inferred clusters derived from Louvain, Leiden, or other common clustering algorithms used for scRNA-seq analyses. Once cluster marker gene sets are obtained, they are combined into a single set, and the rankings across this global marker gene set are used to compute pairwise AO distances, which in turn produce a tree that visualizes cluster similarities. This final tree is used to quantify highly similar clusters, where AO z-scores indicate the likelihood that cluster pairs may share the same cell identity (Fig. 1B).

## Benchmarking the average overlap metric in truth-known data

To evaluate AO for quantifying differences between inferred clusters in scRNA-seq analyses, we benchmarked its performance in datasets where cell identity was known from an experimental assay separate from RNA measurements. We used a dataset of pre-sorted populations of blood cells, which we refer to as *Zhengmix8eq* (Duo et al, 2018; Zheng et al, 2017). We performed benchmarking on both the complete dataset with highly heterogenous, well-separated populations like B cells and monocytes, and also on a subset of more homogenous populations exclusively composed of T cells. In addition, we used a CITE-seq dataset of cord blood mononuclear cells (CBMCs) (Stoeckius et al, 2017), in which ground truth cell identities were obtained using surface protein profiles (Fig. EV1A–C).

We benchmarked AO in each dataset by first clustering cells using the Leiden algorithm at various resolutions (1.0–2.0 for *Zhengmix8eq*, 1.5–2.0 for the T-cell subset, and 0.6–1.2 for the CBMCs). We used the union of top differentially expressed genes obtained across different numbers of top marker genes (5, 10, 25, 100, 200, and 500 per cluster) to calculate pairwise AO distances and perform hierarchical clustering of the resulting Leiden clusters. Using these trees, we iteratively merged the clusters with the highest AO z-scores (Fig. 1B) until the number of remaining clusters matched the number of ground truth labels.

To benchmark AO against other metrics utilized for assessing similarities between single-cell clusters, we performed the same hierarchical clustering with pairwise Pearson, Spearman, and Kendall-Tau correlations, as well as Euclidean distance, using the marker genes' normalized expression counts. We also benchmarked the use of these same metrics applied to the first 50 principal components, which summarize expression across all genes. In addition, to provide a baseline for comparison, we calculated the performance of clustering achieved with the direct overlap of the clusters' top marker gene sets, as well as a Leiden clustering by itself without any refinement, with resolution parameters highly tuned to achieve the exact number of ground truth populations (Fig. EV1A–C). We repeated each of these analyses 100 times with a new random seed for initiating the Leiden algorithm.

For performance evaluation, we first quantified the agreement between the derived cell populations and ground truth identities using the adjusted Rand index (ARI) (Fig. 2A,D), adjusted mutual information (AMI), and the Fowlkes–Mallows score (FMS) (Appendix Figs. S2–S4). Across all datasets and initial Leiden resolutions tested, the AO metric applied to cluster marker genes

had the best performance compared to other metrics, whether calculated on expression counts or principal components (Figs. 2A,D and EV2A–C; Appendix Figs. S2–S6). In particular, the best performance was achieved using a fewer number of marker genes (Fig. EV3A–C). In comparison, with the exception of the Pearson correlation applied to the *Zhengmix8eq* T-cell subset (Fig. EV2B,C), all other applications using principal components performed the poorest. The AO metric also showed the least variance in performance across all iterations of benchmarking with different starting Leiden resolutions in all datasets. In fact, the AO metric performed very similarly to the baseline obtained from fixed, highly optimized Leiden clustering that produced the same number of clusters as ground truth populations.

Second, we tested the hypothesis that each inferred cluster ideally contains a single true cell identity and that its gene expression profile is highly similar to that of its corresponding majority ground truth population. To this end, we calculated cluster purity, defined by the extent to which each cluster contained a single ground truth label, and computed the difference in average gene expression between the clusters and their corresponding ground truth populations. Again, across all datasets and resolutions tested, average cluster purity was the highest, and transcriptional differences were the lowest, using the AO-based trees (Figs. 2B,C,E,F and EV2A–C; Appendix Figs. S5 and S6). As expected, utilizing direct overlap of cluster marker gene sets performed extremely poorly across all datasets when less than 25 marker genes per cluster were included (Appendix Fig. S7). In these scenarios, dissimilar cell populations are not likely to share marker genes. In contrast, when over 100 marker genes per cluster were included, direct overlap and AO showed similar performance in capturing true cell identities in all datasets.

To illustrate the use of AO z-scores for interpreting the significance of AO similarity, we show pairwise AO z-scores calculated for both the ground truth populations and a Leiden clustering of the *Zhengmix8eq* dataset, performed at a previously benchmarked resolution of 1.0, using each population's top 25 marker genes (Fig. EV4A,B). Pairwise AO z-scores between the most dissimilar cell types, such as B cells, monocytes, and natural killer cells, correspond to small or negative values. In contrast, pairwise AO z-scores for highly similar T-cell populations are positive, ranging from 5.4 to 14.7 standard deviations (Fig. EV4C,D). As expected, in high-resolution clustering where highly similar cell populations may split into multiple groups, the corresponding pairwise AO z-scores become exceedingly large, indicating that greater AO z-scores consistently point to more similar cell identities.

Put together, AO robustly and reproducibly measures similarities between single-cell clusters based on the ranking of marker genes and significantly outperforms correlation-based and Euclidean distance metrics in identifying and characterizing populations present in diverse hematopoietic cellular populations.

## Thymic T-cell development at single-cell resolution

Specific stages of T-cell development in the thymus have been characterized with single-cell transcriptomic studies, starting from double-negative (DN) populations DN1-DN4, advancing to immature single positive (ISP) and double positive (DP), and eventually mature CD4 and CD8 T cells. While the T-cell development trajectory in the

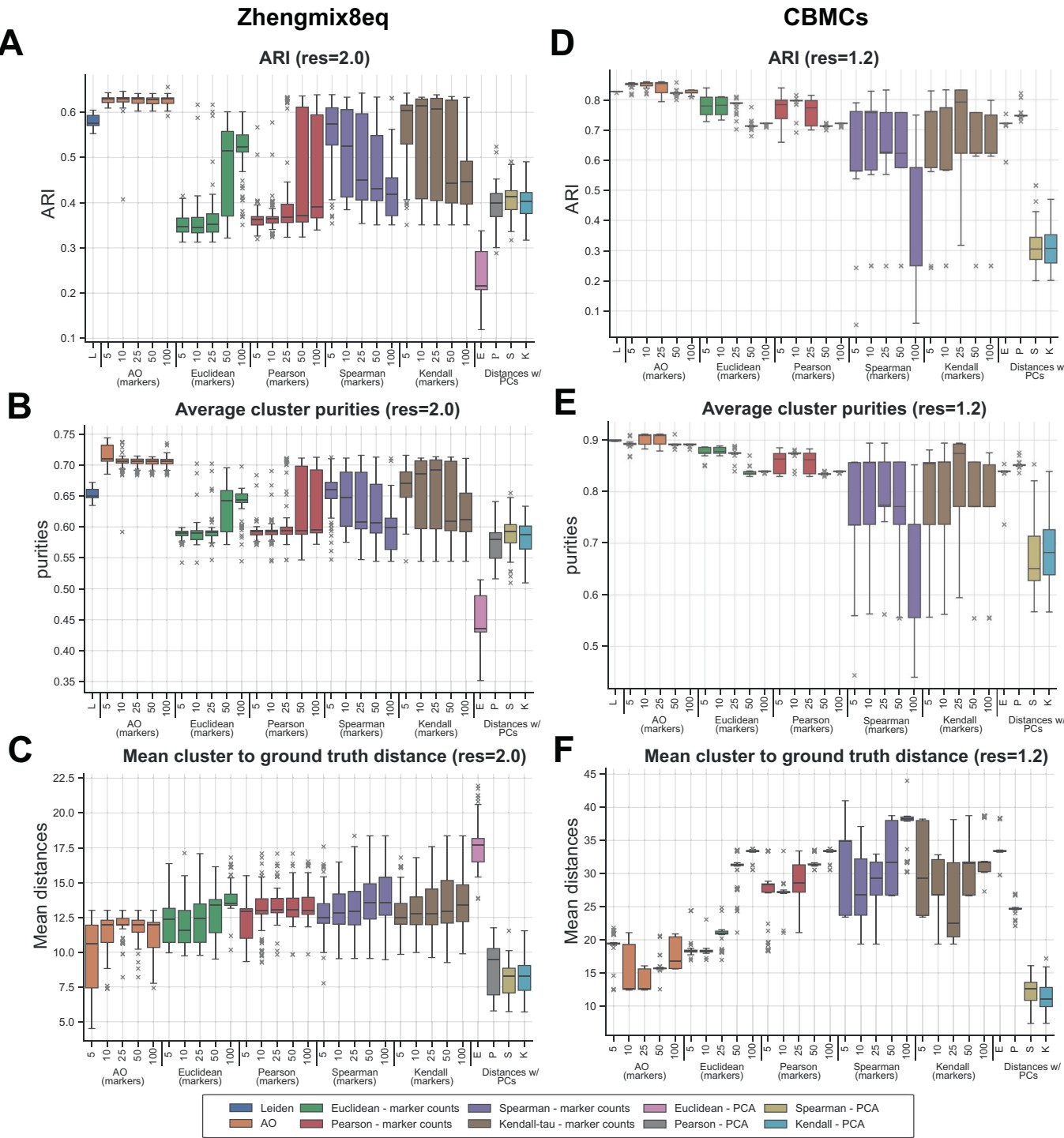

**Figure 2. Benchmarking average overlap versus other distance metrics used in hierarchical clustering of single-cell clusters.**

(A–C) Distributions of adjusted Rand Index (ARI) (A), average cluster purity (B), and mean distances between inferred merged clusters and their corresponding ground truth populations (C) across different metrics and number of marker genes used in the *Zhengmix8eq* dataset. 8 final merged clusters were formed from an initial Leiden clustering with resolution 2.0. (D–F) Distributions of adjusted Rand Index (ARI) (D), average cluster purity (E), and mean distances between inferred merged clusters and their corresponding ground truth populations (F) across different metrics and number of marker genes used for the CBMC CITE-seq dataset. Seven final merged clusters were formed from an initial Leiden clustering with resolution 1.2. Data information: Hierarchical clustering was performed with each combination of metrics and/or number of cluster marker genes 100 times. In all boxplots (A–F), the midpoints represent median values. The bounds of the box correspond to the 25th and 75th percentile values (Q1 and Q3, respectively). The whiskers of the plots extend to points that lie within 1.5 IQRs of the lower and upper quartiles Q1 and Q3, where IQR = Q3 – Q1 is the interquartile range. Observations that fall outside this range are displayed independently.

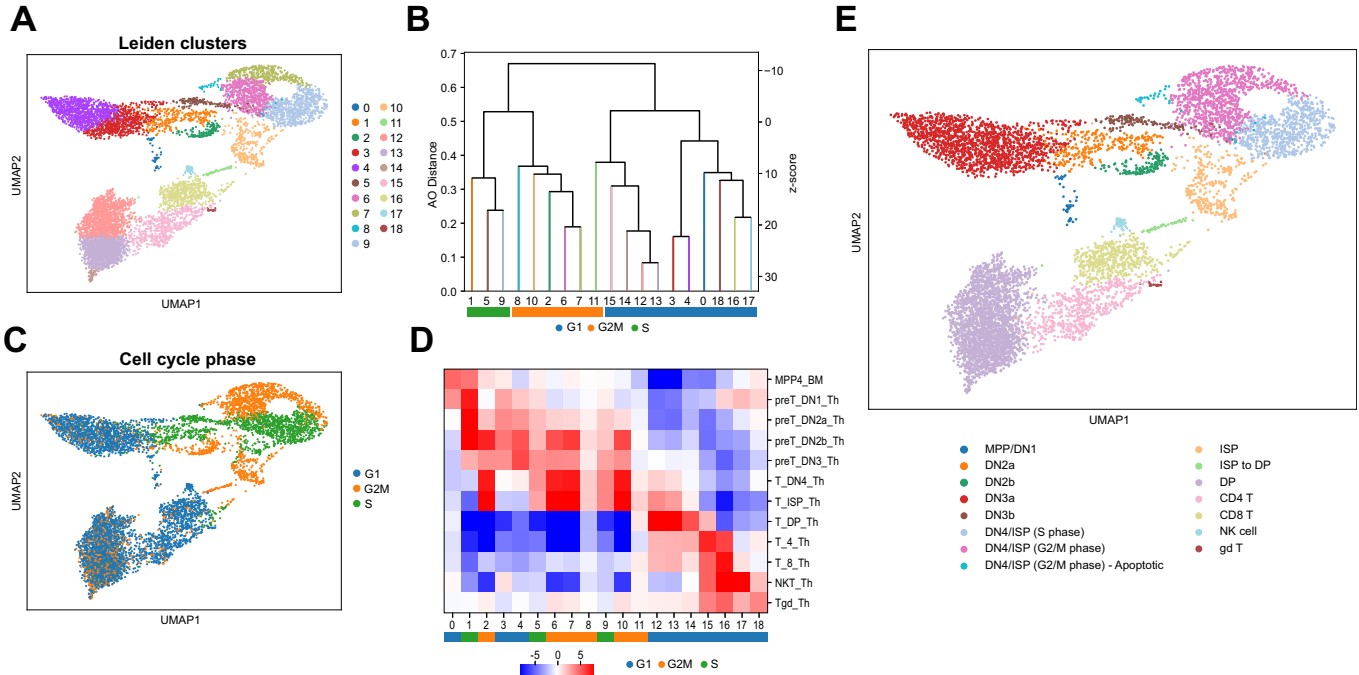

**Figure 3. Utilizing average overlap to characterize stages of T-cell development in a mouse thymus.**

(**A**) A UMAP plot of mouse thymocytes showing 19 inferred clusters obtained using the Leiden algorithm. (**B**) Cluster tree generated from hierarchical clustering of Leiden clusters with average overlap of marker gene rankings, based on the combined set of each cluster's top 25 marker genes. Clusters are largely grouped based on the differential expression of genes related to the cell cycle. (**C**) A UMAP plot of the mouse thymocytes, where each cell is annotated with its inferred cell cycle phase. (**D**) A heatmap summarizing the expression of Leiden cluster marker genes in sorted bulk populations of thymocytes from ImmGen. Each cluster is also colored with its inferred cell cycle phase. (**E**) Final annotations of developing mouse thymocytes.

thymus has been partially resolved in normal and diseased states, characterizing double-negative cell subpopulations using single-cell approaches has been especially challenging, in particular when measuring total thymocytes without prior cell sorting, leading to grouping of all DN subpopulations together without specifically differentiating DN1-DN4 states (Belver et al, 2019; Oh et al, 2023a; Park et al, 2020; Tottone et al, 2021).

To address this challenge and to demonstrate the utility of the AO metric in elucidating the relationships between thymic cell populations, we used CANTAO to analyze single-cell transcriptomic data previously published from mouse thymocytes (Tottone et al, 2021). We obtained 19 single-cell clusters at default Leiden resolution of 1.0 (Fig. 3A), and tested the hypothesis that T-cell clusters with high AO similarity correspond to cell populations with similar identity and state (Fig. 3B,C; Appendix Fig. S8). We then performed hierarchical clustering using pairwise AO distances between each cluster's rankings of marker genes. We utilized the MyGeneSet tool, available as a part of the Immunological Genome Project (ImmGen) (Heng et al, 2008) to guide annotating T-cell identities for unsupervised clusters, and mapped the expression of each cluster's top 50 marker genes across 12 sorted mouse thymic cell populations using composite z-scores (Fig. 3D). This analysis linked the expression profiles of the 19 de novo single-cell clusters to MPP4 (a subset of multipotent progenitors from the bone marrow), DN1, DN2a, DN2b, DN3, DN4, ISP, DP, mature CD4 and CD8, natural killer, and γδ T cells in ImmGen cell populations (Yoshida et al, 2019).

## The cell cycle characterizes thymic single-cell clusters

Marker genes in multiple single-cell clusters included those related to phases of the cell cycle. Hierarchical clustering based on AO of marker gene sets identified distinct groups of single-cell clusters; therefore, we hypothesized that similarity in the rankings of differentially expressed genes, as quantified by AO, would group the clusters based on cell cycle phases. We tested this hypothesis by scoring each cell based on the expression of known cell cycle genes and predicting their cell cycle phase (Tirosh et al, 2016) (Fig. 3B). We found that clusters 1, 5, and 9 were enriched with cells in the S phase while clusters 2, 6, 7, 8, 10, and 11 were enriched with cells in the G2/M phase (Fig. 3C). These groups of clusters also showed very high AO similarity in their marker gene sets, with clusters 1, 5, and 9 in the S phase corresponding to AO z-scores between 11.5 and 19.5. Meanwhile, AO z-scores for clusters 2, 6, 7, 8, 10, and 11, which were in the G2/M phase, ranged from 10.0 to 23.1 (Appendix Fig. S8). The remaining clusters were enriched with cells in the G1 phase, split into two groups of clusters 0, 3, 4, 16, 17, and 18 plus clusters 12, 13, 14, and 15 (Fig. 3C). These marker genes associated with the cell cycle were also highly expressed in multiple ImmGen bulk populations, including the DN2b, DN4, and ISP populations (Fig. 3D). These results suggest that the cluster groupings that emerge from AO clustering reflect cell cycle signatures, pointing to stages of T-cell development involving rapid expansion and proliferation, such as those that occur after T-lineage commitment and β-selection (Oh et al, 2023b).

## Annotation of thymic single-cell clusters during development

Using the annotations from MyGeneSet and Immgen sorted populations, we found that the earliest thymocyte progenitors (MPP4 and DN1 cells) mapped most strongly to cluster 0, which had marker genes that did not show significant expression in any other ImmGen thymic population. At the same time, cluster 0 did not show high similarity to any other cluster according to AO; thus, we annotated cluster 0 as a mix of multipotent progenitors (MPP) and DN1 cells.

Cluster 1 corresponded strongly to DN2a populations, and cluster 2 was inferred as DN2b cells based on its marker genes' expression specifically in the DN2a/b and earlier progenitor populations. Marker genes for cluster 2 were also highly expressed in proliferating DN4 and ISP populations, stemming from a 36% overlap with known cell cycle genes, including *Hmmr* and *Nusap1*.

Clusters 3 and 4 marker genes were highly expressed in DN3 populations and showed marked AO similarity (z-score = 25.3). In addition, marker genes for cluster 5 showed relatively high expression in DN3 cells. For deeper characterization of these cells into DN3a and DN3b populations, we scored cells in clusters 3, 4, and 5 against genes upregulated in purified wild-type DN3a and 3b cells (Vogel et al, 2016). While cluster 4 highly correlated with DN3a, and cluster 3 modestly correlated with both phases, cluster 5, which is in the S phase of the cell cycle, showed similarity to DN3b cells, together indicating a transition of cells from DN3a to DN3b states (Appendix Fig. S9). Accordingly, we merged clusters 3 and 4 together to form a DN3a population, and annotated cluster 5 as the DN3b population.

Clusters 6, 7, 8, and 9 mapped strongly to ImmGen DN4 and ISP populations. While cluster 9 was distinguished as DN4/ISP cells in the S phase of the cell cycle, clusters 6, 7, and 8 grouped together based on their pairwise AO and association with the G2/M phase of the cell cycle. Of note is cluster 8, which contained cells with low expression of a select set of genes that were not expressed in any other clusters (Appendix Fig. S10). Relative to cells in clusters 6 and 7, cells in cluster 8 showed an upregulation of mt-Co1, mt-Co2, mt-Co3, and other mitochondrial genes. In fact, mitochondrial genes made up for on average 7.8% of this cluster's expressed transcripts, indicating that a portion of cluster 8 would fall just under the 12% mitochondrial expression threshold we used for filtering low-quality cells at the start of analysis. Put together, we interpreted cluster 8 as representing apoptotic DN4/ISP cells (Appendix Fig. S11).

Cluster 10 showed the highest correspondence to ImmGen ISPs, while cluster 11 was inferred as an intermediate population between the ISP and DP stages based on its marker genes' expression in late development T cells (Appendix Fig. S10). Clusters 12, 13, and 14 were identified as parts of a larger DP population, as they each mapped strongly to ImmGen DP cells and showed the highest AO similarities across all clusters (z-score range: 24.1–31.1). The maturing stages of T-cell development were represented by cells in clusters 15 as single positive CD4 and cluster 16 as single positive CD8 T cells. Finally, clusters 17 and 18 were assigned to smaller groups of natural killer cells and γδ T cells, respectively. Thus, with the use of CANTAO and marker gene expression in ImmGen sorted populations, we were able to specifically annotate stages of T-cell development in the thymus, most notably the intermediate stages of double-negative development (Fig. 3E).

## Orthogonal analysis of thymic single-cell clusters during development

Lastly, we explored characterizing the stages of T-cell development in our data using orthogonal methods. These analyses were performed downstream of the original Leiden clustering. First, we asked whether pseudotime inference could recover cell identities in these developing T cells and used diffusion pseudotime (Haghverdi et al, 2016) specifying cluster 0 as the starting point. The inferred trajectory matched our annotated stages of development and supported a later developmental trajectory for cells in clusters 6 and 7 compared to cells in cluster 2; hence, partially distinguishing DN2b from DN4/ISP cells despite being in the same cell cycle phase (Appendix Fig. S12A). This pattern, however, did not hold when cluster 2 was chosen as the analysis's starting point (Appendix Fig. S12B). Cells in clusters 6 and 7 were assigned the same starting pseudotime value of 0 as the chosen cluster 2, indicating that pseudotime calculations can be confounded by strong cell cycle signatures.

Second, we asked whether any of the previously benchmarked metrics (i.e., Pearson, Spearman, and Kendall-Tau correlations, or Euclidean distance) applied to the expression counts of the marker genes could infer T-cell similarities. This analysis showed that the resulting hierarchical clustering trees did not consistently group the single-cell clusters based on phases of cell cycle or stages of T-cell development (Appendix Fig. S13), highlighting the strength of AO in inferring biologically meaningful relationships by using the rankings of differentially expressed genes as opposed to their counts, especially in the context of highly similar cells. Finally, we performed an automatic annotation of each individual cell using the singleR tool (Aran et al, 2019), with sorted bulk-RNAseq populations of mouse thymocytes in ImmGen as the reference for annotation (Appendix Fig. S14). These algorithmically assigned labels aligned closely with CANTAO's unsupervised approach and confirmed the accuracy of AO-based clustering followed by annotation; however, they lacked the resolution to discern the cell cycle signature that is critical to specific stages of thymocyte development.

Given the prominent role of the cell cycle in thymus development, we tested whether regressing out the cell cycle effects would impact cluster annotation and inference. As expected, while de novo Leiden clustering was able to identify populations in the G1 phase of the cell cycle, the DN2, DN4, and ISP cells, which were previously identified to be in separate phases of the cell cycle, were intermixed and could not be inferred as separate populations (Fig. EV5A–C). This effect could be readily seen when our original annotations that included cell cycle, or the singleR reference annotations, were overlaid on a UMAP produced from regressed gene expression counts (Fig. EV5D,E). Regressing the cell cycle out resulted in lost separation between DN2a/b populations, as well as less clear separation of DN3b versus DN4 and ISP populations.

Overall, our extended analyses of truth-known hematopoietic populations as well as unsorted mouse thymocytes showed that AO could accurately guide the clustering and annotation of highly homogenous cell populations and help characterize the trajectory of T-cell development in the thymus, including the detection of elusive double-negative (CD4-CD8-) cells. We have implemented CANTAO in Python and have designed a package to work within the Scanpy framework for seamless integration into single-cell analysis workflows.

## Discussion

In this work, we propose using average overlap, a top-weighted metric that quantifies the similarity of ranked lists, to compare clusters in scRNA-seq analysis. Hierarchical clustering using rank- and correlation-based metrics to compare the transcriptomic profiles of inferred clusters in scRNA-seq analysis is not new. Current implementations, including those in Seurat and Scanpy, utilize distances based on gene expression counts or principal components. In contrast, AO measures cluster similarity by relying on marker gene rankings derived from differential gene expression analysis.

We compared AO to correlation-based metrics and Euclidean distance using Leiden clusters derived from biological, ground-truth datasets. We showed that merging unsupervised clusters into groups with the AO metric based on gene rankings resulted in cell labels that corresponded to ground truth populations more accurately than other tested methods. AO was also far more consistent in its performance, and over many Leiden clustering iterations, produced merged populations that were biologically meaningful and had highly similar gene expression profiles to those of the original ground truth populations. Expression counts across highly similar populations are highly correlated, irrespective of the feature selection procedure and the amount of cluster marker genes used. This high transcriptional correlation often results in inconsistencies that make distinguishing subtle differences between subpopulations a challenge. Even utilizing a simple Euclidean distance across cluster marker genes, or computing distances using principal components which effectively summarize gene expression, did not result in competitive performance. Meanwhile, restricting analysis to just gene rankings based on differential expression in our implementation of AO allowed for robust measurement of subtle transcriptional differences between single-cell clusters.

Moreover, AO performed well compared to other distance metrics due to its top-weighted property. While performance for all metrics either diminished or became inconsistent as more marker genes were included, AO-based hierarchical clustering was minimally affected, independent of the number of marker genes used. Reincorporating information about gene expression variation by way of utilizing principal coordinates recovered the performance or reduced its variability for Pearson correlation or Euclidean distance to an extent; however, this was not consistent across all datasets and resolutions. We interpret this observation as evidence for the effect commonly known as the curse of dimensionality (Bellman and Rand Corporation, 1957), where the differences in distances between clusters become increasingly similar as a higher number of marker genes are added. In contrast, differences at the top of rankings as measured by AO, correspond to the most differentially expressed genes, and reflect cellular differences more precisely than the genes at the bottom of the rankings. As such, the AO metric is generally not as sensitive to the curse of dimensionality as the other benchmarked metrics, as demonstrated by its top performance with extremely low variability.

CANTAO can quantify subtle differences in otherwise highly transcriptionally correlated cell populations. In our thymus data, AO-based hierarchical clustering revealed cell cycle genes as an important part of cluster marker gene similarity, indicating stages of thymocyte development that involve a high degree of proliferation after passing particular checkpoints. However, showing high correlation with ImmGen reference expression profiles was not sufficient to characterize cell cycle signatures in single-cell clusters that corresponded to DN2b, DN4, and ISP cells at once. By measuring similarity in differential gene rankings, CANTAO grouped clusters based on inferred cell cycle phases with great specificity and guided the identification of stages of T-cell development involving increased expansion and proliferation.

While cell cycle variation in atlas-type analyses that include many heterogenous, well-separated cell populations can be regressed out to identify the cells belonging to the same types in differing cell cycle phases, the proliferative states inferred from single-cell expression data are in fact important identifying features of developing thymocytes and directly inform cluster separation when using the Leiden algorithm. When we attempted to regress the cell cycle signature out, the cluster separation was noticeably lost, except for cells in the G1 phase. This was most notable in the DN2 cells, where the DN2a/b populations, which previously separated due to their different cell cycle phases, formed a homogenous population. The DN4 and ISP cells also became intermixed into a single large population, which included the DN3b population primarily in the S phase. When we overlayed cell cycle scores and singleR automated annotations on the projection of cells after regressing cell cycle out, we noticed that all DN2, DN4, and ISP cells still coincided with the G2M and S phases, but not the G1 phase. As there are no non-proliferative states of these populations of thymocytes according to both our analysis and automatic annotation, we then conclude that the differential expression of cell cycle genes in combination with the differential expression of known marker genes of T-cell development can uniquely help in the identification of specific stages of double-negative cells when using single-cell RNA data alone. More broadly, our results offer a cautionary tale about removing cell cycle signatures by default from general single-cell analyses.

In addition, AO helped to characterize a small subset of cells in the G2/M phase as a population of apoptotic DN4/ISP. Cells in this cluster were retained simply due to the choice of mitochondrial expression threshold used to account for low-quality cells, yet showed relatively high correspondence to ImmGen DN4 and ISP populations, supporting their biological relevance. Thymocytes may undergo positive selection based on their ability to bind to MHC ligands; thymocytes unable to bind MHC undergo death by neglect (Baldwin et al, 2004). The combination of relatively high mitochondrial gene expression which typically indicates dying cells, high correspondence to bulk DN4 and ISP populations in ImmGen, and the detection and quantification of unique marker genes via the Leiden clustering algorithm and the AO metric supports this interpretation. Altogether, unique subpopulations of thymocytes characterized with the aid of AO may be missed when relying solely on reference transcriptomic profiles.

The analysis and comparison of cluster marker genes are necessary steps in uncovering novel cell populations. As we have demonstrated with our analysis of thymus data, one clustering resolution may not yield clusters that simultaneously represent broad well-defined populations as well as smaller novel subpopulations. Moreover, while some clusters may separate due to existing biological variations, others may arise due to noisy clustering. AO-measured distances based on the ranking of differentially expressed genes provide a quantitative method for evaluating transcriptional similarity independent of clustering algorithms and parameters

used. In the context of differentiating cells, alternative methods such as pseudotime inference, which calculates the relative position of a cell across gene expression gradients, can theoretically aid cluster annotation (Haghverdi et al, 2016; Street et al, 2018; Trapnell et al, 2014). However, pseudotime algorithms rely heavily on the choice of starting population, and as we showed in an application to our thymus data, they may group cells in the same cell cycle phase despite transcriptional correspondence to different biological populations. In addition, when there are groups of many distinct cells in general, a trajectory cannot be inferred altogether, and a quantitative method for cluster comparison is still needed.

In conclusion, we propose using CANTAO and the underlying average overlap metric for direct quantification of marker gene similarity with broad applicability in diverse biological settings suited to explore and clarify the heterogeneity of cells in most challenging contexts that arise from differentiating cells or involve clonal populations, such as those driving cancer progression.

# Methods

**Reagents and tools table**

| Reagent/resource | Reference or source | Identifier or catalog number |
| --- | --- | --- |
| **Experimental models** | | |
| **Recombinant DNA** | | |
| **Antibodies** | | |
| **Oligonucleotides and other sequence-based reagents** | | |
| **Chemicals, enzymes, and other reagents** | | |
| **Software** | | |
| Python | https://www.python.org/ | |
| **Other** | | |

## The sc_average_overlap Python package

We have developed CANTAO and its implementation of the average overlap metric in Python and have published the "sc_average_overlap" package (https://github.com/chrisvthai/sc_average_overlap), designed to work seamlessly with Scanpy, a commonly used Python library for scRNA-seq analyses. The package includes functions for computing pairwise AO scores between two clusters, performing hierarchical clustering of cell populations, and storing the resulting dendrogram in Scanpy's native AnnData object. For hierarchical clustering, the AO values are subtracted from one to generate a distance metric, since AO ranges between 0 and 1, with higher overlap scores meaning a lower distance, and vice versa.

## Benchmarking the average overlap's performance

To benchmark AO when used as the metric for hierarchical clustering of cell populations in single-cell RNA-seq data, we

utilized the *Zhengmix8eq* dataset, generated for the purpose of benchmarking clustering performance in scRNA-seq analysis (Duo et al, 2018). We used both the complete dataset, as well as a subset consisting of only five T-cell populations, which were CD8 cytotoxic cells and CD4 populations, including helper, memory, naïve, and regulatory cells. In addition, we used a dataset of cord blood mononuclear cells published for the CITE-seq method (Stoeckius et al, 2017). To derive ground truth labels for the CITE-seq data, we clustered the data using only the counts of surface proteins (ADTs), excluding *CCR5*, *CCR7*, and *CD10*, due to their generally low expression across most cells. We also removed cells whose RNA-expression profiles consisted of less than 90% human genes. We performed PCA, and constructed a k-nearest neighbors graph using each cell's 15 nearest neighbors. Finally, we clustered the data using the Leiden algorithm at resolution 0.15, and additionally filtered out clusters containing potential doublets based on high enrichment of multiple surface proteins corresponding to completely different cell types.

We processed all scRNA-seq data using Piccolo (Singh and Khiabanian, 2024) to perform feature selection and normalization, selecting 3000 highly variable genes for downstream analysis. We then performed Principal Component Analysis (PCA) and Leiden clustering with a resolution of 2.0 using Scanpy. We performed this process and all future downstream analysis 100 times in total, each with a different random seed given to the Leiden algorithm.

We generated marker genes for each resulting single-cell cluster by performing differential expression analysis using the Wilcoxon rank-sum test and comparing each cluster's gene expression versus the rest of the cells, ranked by significance. We then grouped single-cell clusters using hierarchical clustering. The metrics used for the hierarchical clustering included Pearson correlation, Spearman correlation, Kendall-Tau coefficient, Euclidean distance, and AO, and were calculated with either the set of all cluster marker genes, composed of the union of each unsupervised cluster's top 5, 10, 25, 50, 100, 200, or 500 differentially expressed marker genes, or the values of the first 50 principal components. When using the set of all cluster marker genes, Pearson, Spearman, and Kendall-Tau coefficients were calculated using each cluster's expression values for each gene, while AO was calculated using the rankings of the cluster marker genes. When using principal component values, only Pearson, Spearman, and Kendall-Tau coefficients were calculated. As additional baselines for comparison, we computed a highly tuned Leiden clustering in each dataset which produced the same number of clusters as the number of ground truth populations, and measured the performance of this partitioning. These resolutions were 0.35, 0.775, and 0.13 for the *Zhengmix8eq* dataset, its T-cell subset, and the CBMC dataset, respectively. We also performed hierarchical clustering of Leiden clusters using direct overlap of marker gene sets as the underlying metric, and calculated the performance of merged clusters using this tree.

Once hierarchical clustering was computed, the clusters were iteratively merged in an automated manner, guided by the pair of clusters with the highest AO similarity, until the number of remaining clusters matched the number of ground truth labels. Performance was evaluated in two ways. First, using the ground truth labels, we calculated adjusted Rand index (ARI), adjusted mutual information (AMI), and the Fowlkes–Mallows score (FMS). Second, we calculated the average cluster purity, defined by the extent to which each cluster contained a single ground truth label.

For the final merged clusters, we calculated the Euclidean distance of average gene expression profiles between the clusters and their corresponding ground truth populations. These distances were calculated using log-normed expression counts scaled to unit variance and a zero mean in all genes.

## Thymocyte analysis using average overlap

We applied CANTAO to a previously published dataset of mouse thymocytes (Tottone et al, 2021). We first performed feature selection and normalization using Piccolo, excluding cells expressing more than 12% mitochondrial genes or 80% ribosomal genes and selecting 3,000 highly variable genes for downstream analysis. We performed PCA and Leiden clustering at resolution 1.0 using Scanpy, and visualized the single-cell clusters using Uniform Manifold Approximation and Projection (UMAP). We performed differential gene expression analysis using the Wilcoxon rank-sum test and assigned the top 25 differentially expressed genes, ranked by significance, as a set of marker genes in each cluster. As we were interested in T-cell populations only, we excluded small populations of granulocytes and B cells detected through scoring against known markers of each cell type. We combined all clusters' marker genes into a single set for calculating AO for subsequent calculation of pair-wise distances between the clusters and hierarchical clustering.

We obtained and visualized the expression of each set of cluster marker genes in 12 sorted populations of thymic cells in mice from the ImmGen project using the MyGeneSet tool (Heng et al, 2008). To summarize the expression of each single-cell cluster's marker genes in ImmGen sorted bulk populations, we first transformed the expression counts into a normalized z-score and then combined the transformed values into a single composite z-score using Stouffer's method.

We used Scanpy's score_genes function for scoring each cell according to phases of the cell cycle using a list of 97 known cell cycle marker genes (Tirosh et al, 2016). We used the same scoring method to distinguish between DN3a and DN3b populations based on a list of genes upregulated in purified wild-type DN3a/b cells (Vogel et al, 2016). We performed cell cycle regression using Scanpy's sc.tl.regress_out function with calculated cell cycle scores.

For automatic annotation of cell identity for thymocytes, we used the singleR tool in conjunction with counts of purified mouse thymocytes from ImmGen as a reference dataset.

## Data availability

A complete package implementing CANTAO is found at https://github.com/chrisvthai/sc_average_overlap. Example Python notebooks that employ the package, including one that details the analysis of developing thymocytes, are provided as well.

The source data of this paper are collected in the following database record: biostudies:S-SCDT-10_1038-S44320-025-00176-4.

## Peer review information

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

## Acknowledgements

This work was supported by the National Institutes of Health (R01CA236936 and R01CA285513), the V Foundation (T2019-012 and T2023-024), Blood Cancer United, formerly The Leukemia & Lymphoma Society (Scholar Award 1386-23), the New Jersey Commission on Cancer Research (COCR23PRG006), and Rutgers Cancer Institute of New Jersey Biomedical Informatics Shared Resource (P30CA072720-5917). CT was a fellow of the Biotechnology Training Program at Rutgers University (NIH T32 GM135141). AS was supported by the New Jersey Commission on Cancer Research (COCR24PDF015).

## Author contributions

**Christopher Thai**: Data curation; Software; Formal analysis; Methodology; Writing—original draft; Writing—review and editing. **Amartya Singh**: Formal analysis; Methodology; Writing—review and editing. **Daniel Herranz**: Conceptualization; Resources; Supervision; Funding acquisition; Writing—original draft; Project administration; Writing—review and editing. **Hossein Khiabanian**: Conceptualization; Resources; Formal analysis; Supervision; Funding acquisition; Writing—original draft; Project administration; Writing—review and editing.

Source data underlying figure panels in this paper may have individual authorship assigned. Where available, figure panel/source data authorship is listed in the following database record: biostudies:S-SCDT-10_1038-S44320-025-00176-4.

## Disclosure and competing interests statement

CT, AS, and DH declare no competing interests. HK is a full-time employee of Regeneron Pharmaceuticals.

# Expanded View Figures

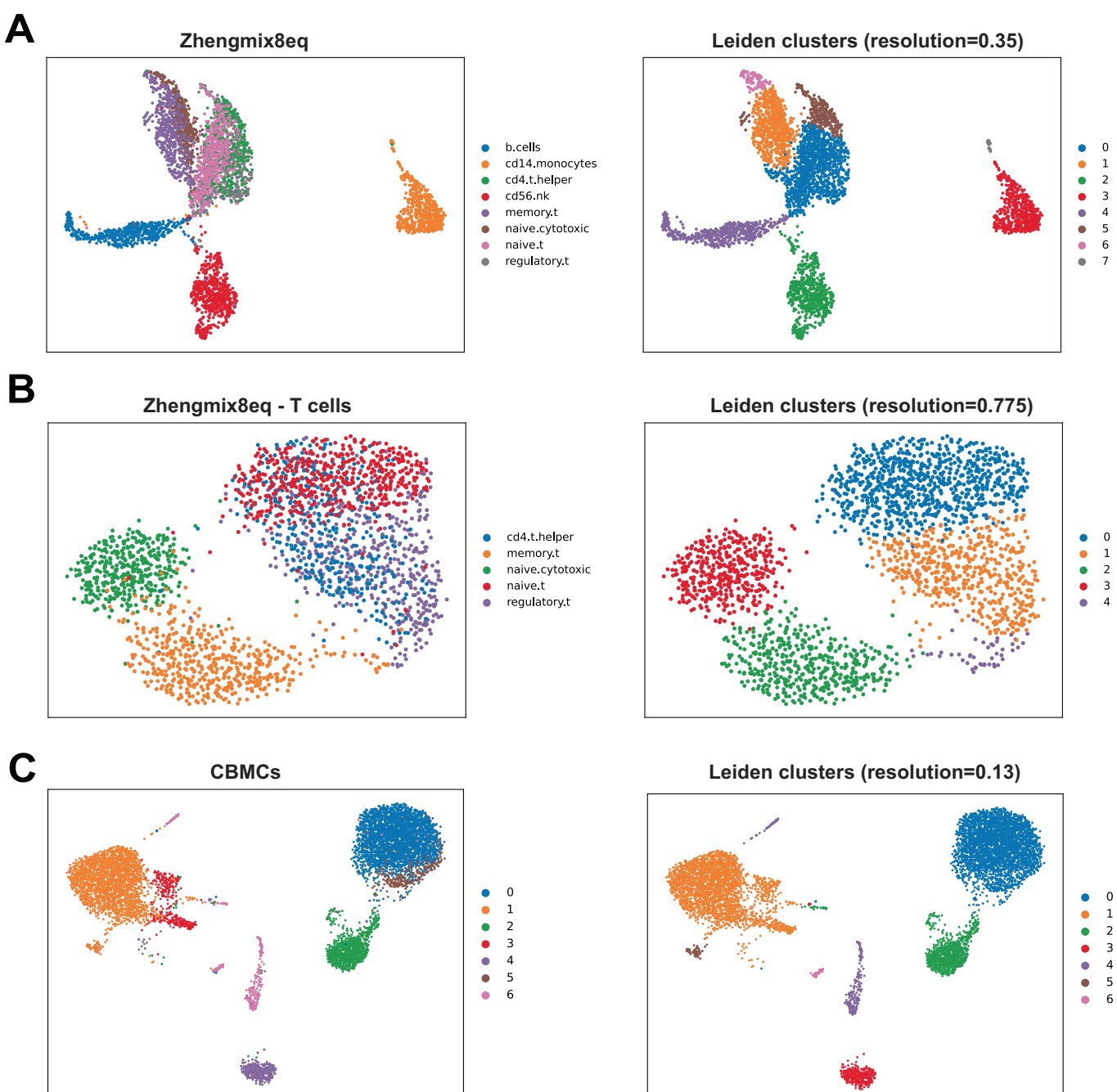

**Figure EV1.   UMAP projections of all datasets used in benchmarking with true labels and fine-tuned Leiden clusters.**

(**A**) UMAP Projections for all cells in the *Zhengmix8eq* dataset. (**B**) UMAP projections of T cells in *Zhengmix8eq*. (**C**) UMAP projections of cells in CBMC dataset, produced from just RNA counts.

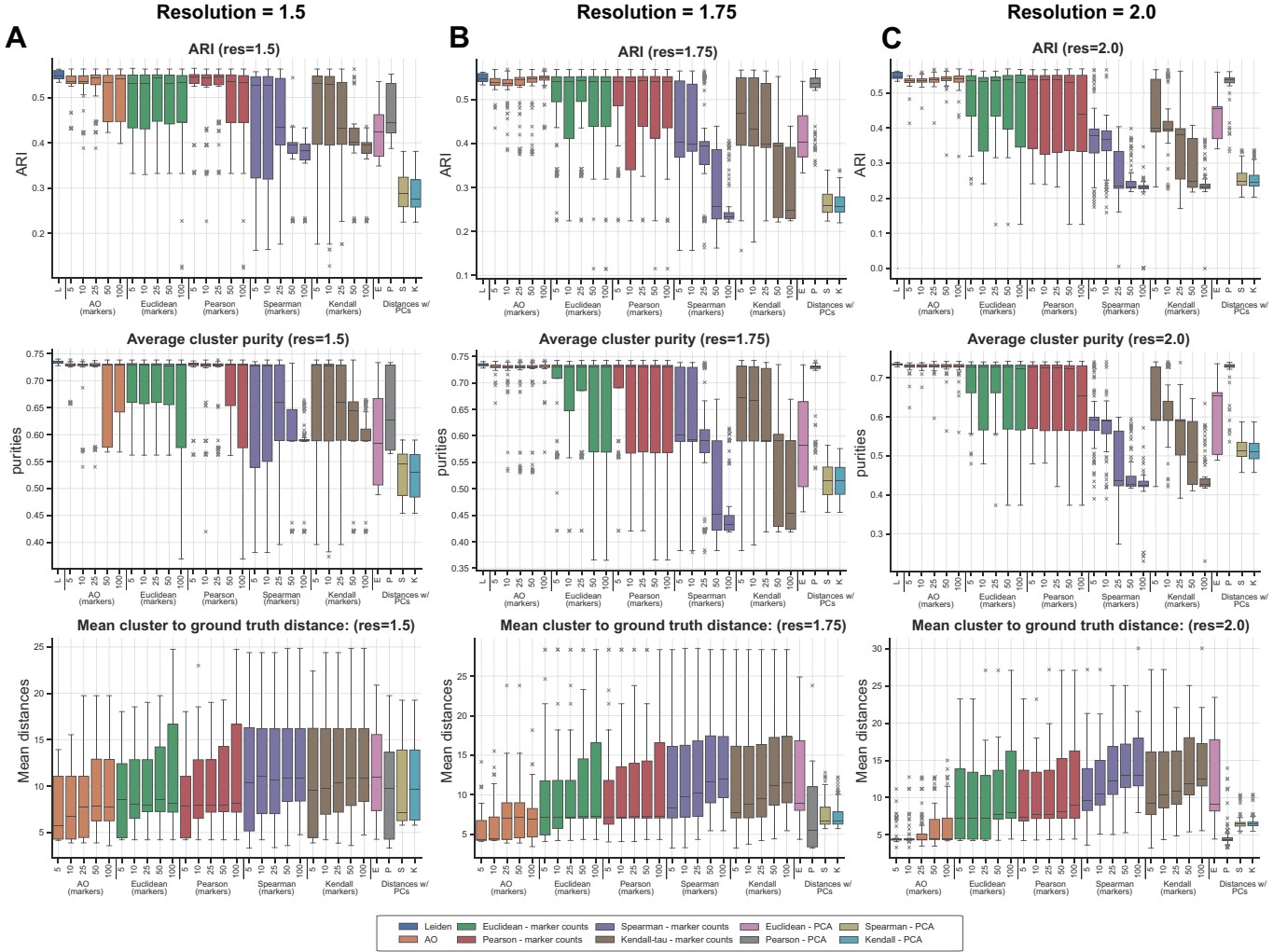

**Figure EV2.    Benchmarking average overlap versus other distance metrics used in hierarchical clustering of single-cell clusters for *Zhengmix8eq* T cells.**

(**A–C**) Adjusted Rand Index (ARI), average cluster purity, and mean distances between inferred merged clusters and their corresponding ground truth populations, based on ground truth labels for 5 cell populations, across different metrics and number of marker genes used, in the T-cell subset of the *Zhengmix8eq* dataset. 5 final merged clusters were produced from initial Leiden clustering at resolutions 1.5 (**A**), 1.75 (**B**), and 2.0 (**C**). Data information: Hierarchical clustering was performed with each combination of metrics and/or number of cluster marker genes 100 times. In all boxplots (**A–C**), the midpoints represent median values. The bounds of the box correspond to the 25th and 75th percentile values (Q1 and Q3, respectively). The whiskers of the plots extend to points that lie within 1.5 IQRs of the lower and upper quartiles Q1 and Q3, where IQR = Q3 – Q1 is the interquartile range. Observations that fall outside this range are displayed independently.

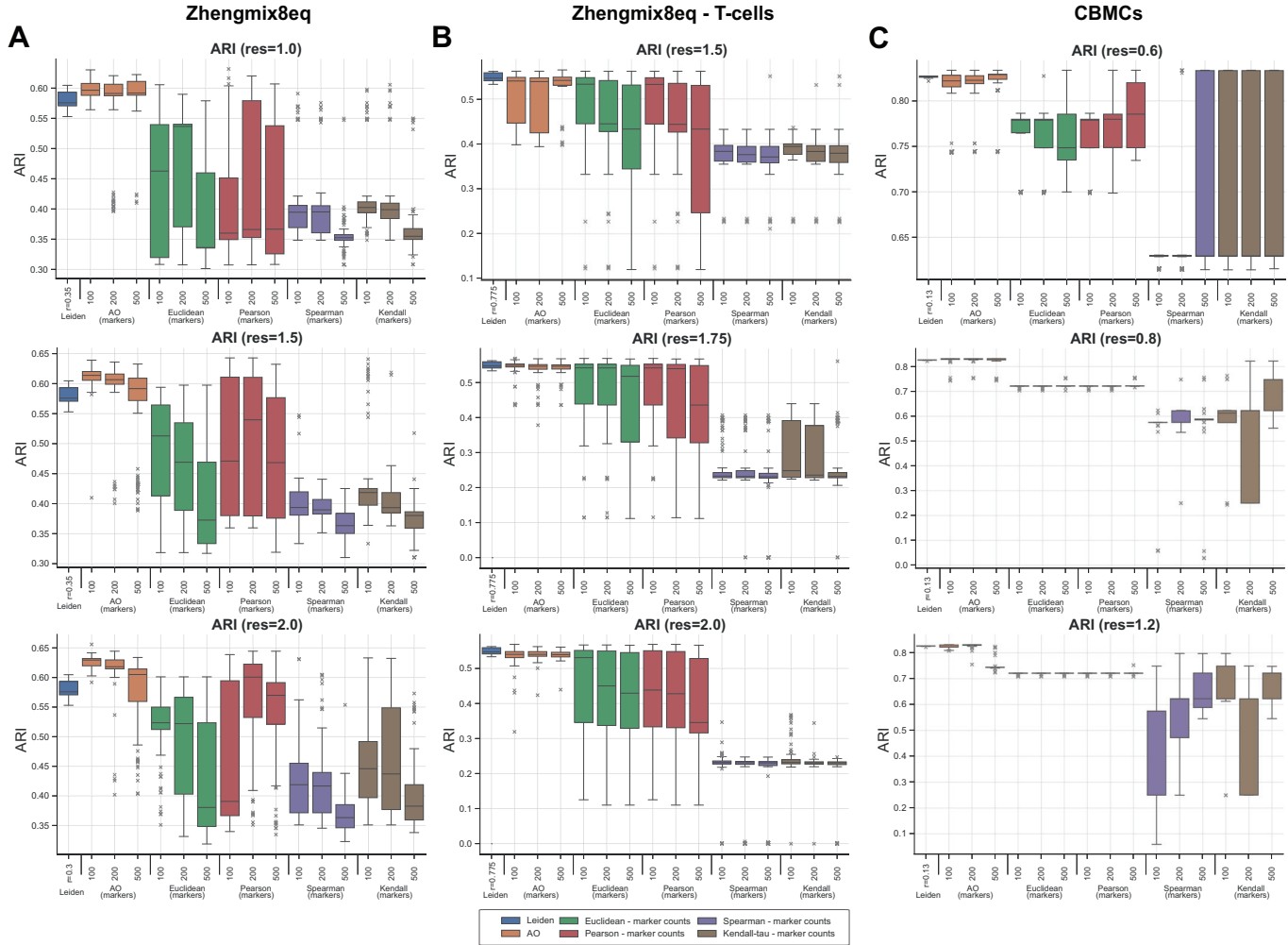

**Figure EV3. ARI in *Zhengmix8eq*, its T-cell subset, and CBMCs when utilizing a high number of cluster marker genes.**

(A–C) ARI of all benchmarks in *Zhengmix8eq* (A), the T-cell subset in *Zhengmix8eq* (B), and CBMCs (C), for all starting Leiden cluster resolutions tested, when defining marker gene sets of length 100, 200, and 500. For AO, there is no noticeable performance gain from using a higher number of marker genes. Data information: Hierarchical clustering was performed with each combination of metrics and/or number of cluster marker genes 100 times. In all boxplots (A–C), the midpoints represent median values. The bounds of the box correspond to the 25th and 75th percentile values (Q1 and Q3, respectively). The whiskers of the plots extend to points that lie within 1.5 IQRs of the lower and upper quartiles Q1 and Q3, where IQR = Q3 – Q1 is the interquartile range. Observations that fall outside this range are displayed independently.

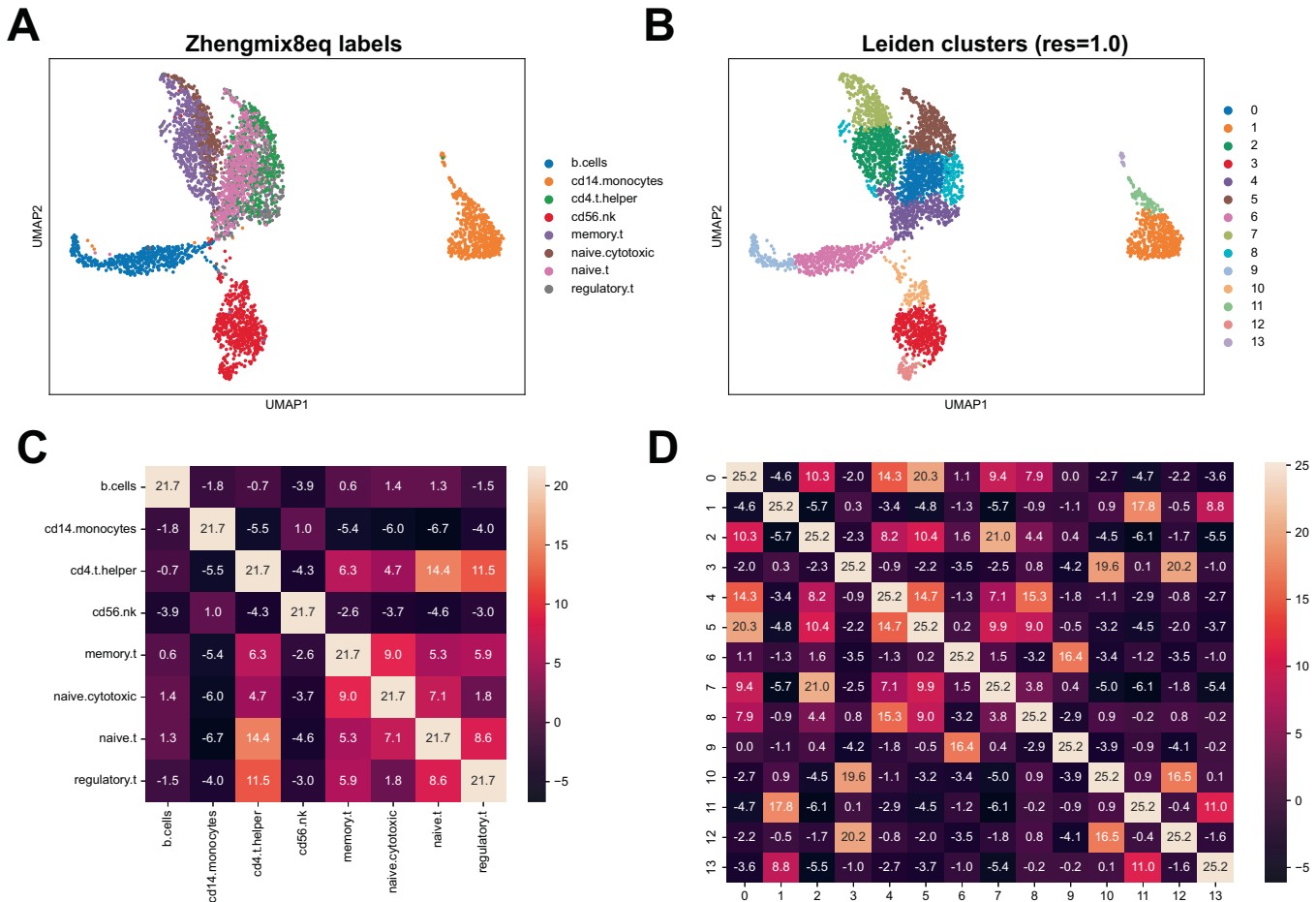

**Figure EV4. Interpreting significance of AO similarity in an example partitioning of the *Zhengmix8eq* dataset.**

(A) UMAP of the *Zhengmix8eq* dataset with true cell labels. (B) A sample Leiden clustering (resolution = 1.0) of the *Zhengmix8eq* dataset projected on the same UMAP. (C) Pairwise AO scores between true cell populations, converted to z-scores. AO was calculated on rankings of a global marker gene set composed of the top 25 differentially expressed genes in each population. (D) Pairwise AO scores between unsupervised Leiden clusters, converted to z-scores, similar to (C).

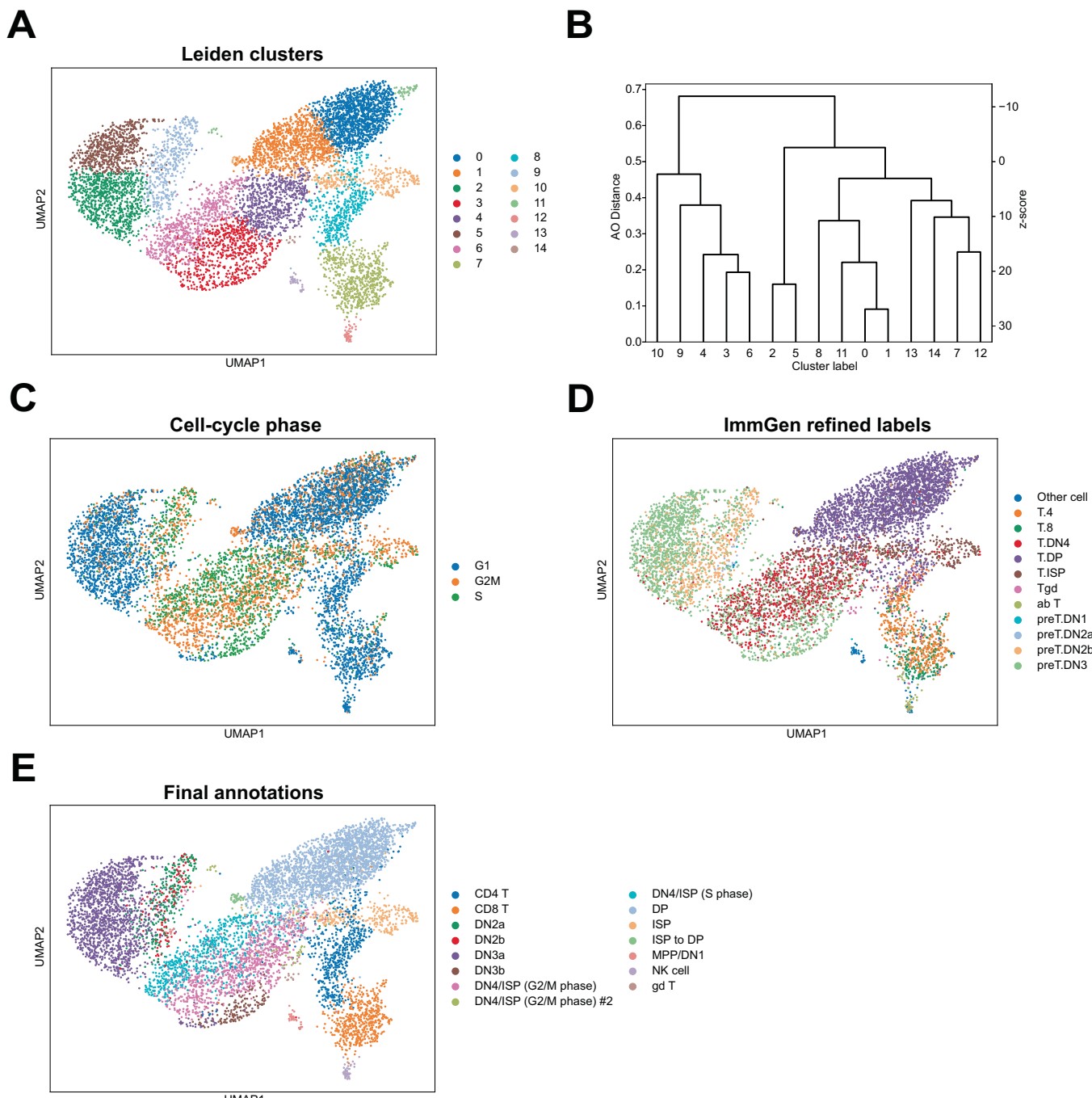

**Figure EV5.  The effect of regressing out cell cycle effects from thymus data.**

(A) UMAP plot of Leiden clustering of thymus data when regressing out cell cycle effects, resulting in 15 populations. (B) AO tree of Leiden clusters shown in (A). (C) Annotated cell cycle phases for each cell, projected on the UMAP produced after regressing out cell cycle. (D) Annotations of individual cells, using the singleR tool with bulk RNA-seq of purified thymocytes in ImmGen as a reference dataset, overlaid on the UMAP produced after regressing out cell cycle. (E) Final annotations of the original analysis of thymus data, overlaid on the UMAP produced after regressing out cell cycle.

