## [Peer Review File · Molecular Systems Biology]

CANTAO: GUIDING CLUSTERING AND ANNOTATION IN SINGLE-CELL RNA SEQUENCING USING AVERAGE OVERLAP

Christopher Thai, Amartya Singh, Daniel Herranz, and Hossein Khiabani

Corresponding author(s): Daniel Herranz (dh710@cinj.rutgers.edu) , Hossein Khiabani (h.khiabani@rutgers.edu)

Review Timeline:

Submission Date:	20th Jun 25
Editorial Decision:	16th Jul 25
Revision Received:	16th Oct 25
Editorial Decision:	19th Nov 25
Revision Received:	21st Nov 25
Accepted:	24th Nov 25

Editor: Jingyi Hou

Transaction Report:

16th Jul 2025

Manuscript Number: MSB-2025-13193-T

Title: GUIDING CLUSTERING AND ANNOTATION IN SINGLE-CELL RNA SEQUENCING USING THE AVERAGE OVERLAP METRIC

Author: Christopher Thai

Amartya Singh

Daniel Herranz

Hossein Khiabani

Dear Dr. Herranz,

Thank you for submitting your work to Molecular Systems Biology. We have now heard back from the two reviewers who agreed to evaluate your manuscript. You will see from the comments below that the reviewers find the general topic to be relevant. However, they raise substantial concerns on your work, which should be convincingly addressed in a major revision of the present manuscript.

We believe the reviewers' recommendations are clearly stated and do not require repetition here. Notably, both reviewers expressed significant concerns about the benchmarking. Reviewer #1 specifically remarked that, without more rigorous benchmarking, the contribution of the proposed AO metric appears incremental. Additionally, both reviewers rated the adequacy of the methodological analysis and validation as "low." Reviewer #1 also assigned a "low" rating to the manuscript's conceptual novelty and overall suitability for publication.

Given these assessments, it is clear that the manuscript is not currently suitable for publication. A thorough and comprehensive response to all reviewer concerns will be essential for any further consideration by the journal. However, addressing these points will likely require substantial additional work, with uncertain outcomes. Considering the scope of the revisions, we are uncertain whether you will be able or willing to undertake them and submit a revised manuscript within the three-month deadline.

That said, given the potential interest of the study, we would be willing to consider a revised manuscript with the understanding that the referee concerns must be fully addressed, and that acceptance of the manuscript would entail a second round of review.

As you may already know, our editorial policy allows in principle a single round of major revision and that, therefore, acceptance or rejection of the manuscript will depend on the completeness of your responses included in the next, final version of the manuscript. For this reason, and to avoid potential frustrations, we would strongly advise against returning an incomplete revision and would also understand your decision if you choose to rather seek rapid publication elsewhere at this stage.

Should you decide to embark in such a revision, revised manuscripts should be submitted within three months of a request for revision; they will otherwise be treated as new submissions, except under exceptional circumstances in which a short extension is obtained from the editor.

Should you find that the requested revisions are not feasible within the constraints outlined here and choose, therefore, to submit your paper elsewhere, we would welcome a message to this effect.

On a more editorial level, we would ask you to address the following issues:

- Please provide a .docx formatted version of the manuscript text (including legends for main figures, EV figures and tables). Please make sure that the changes are highlighted to be clearly visible.
- Please provide individual production quality figure files as .eps, .tif, .jpg (one file per figure).
- Please provide a .docx formatted letter INCLUDING the reviewers' reports and your detailed point-by-point responses to their comments. As part of the EMBO Press transparent editorial process, the point-by-point response is part of the Review Process File (RPF), which will be published alongside your paper.
- Please note that all corresponding authors are required to supply an ORCID ID for their name upon submission of a revised manuscript.
- We replaced Supplementary Information with Expanded View (EV) Figures and Tables that are collapsible/expandable online (see examples in <http://msb.embopress.org/content/11/6/812>). A maximum of 5 EV Figures can be typeset. EV Figures should be cited as 'Figure EV1, Figure EV2' etc... in the text and their respective legends should be included in the main text after the legends of regular figures.

Additional Tables/Datasets should be labeled and referred to as Table EV1, Dataset EV1, etc. Legends have to be provided in a separate tab in case of .xls files. Alternatively, the legend can be supplied as a separate text file (README) and zipped together with the Table/Dataset file.

For the figures and tables that you do NOT wish to display as Expanded View figures, they should be bundled together with their legends in a single PDF file called *Appendix*, which should start with a short Table of Content. Each legend should be below the corresponding Figure/Table in the Appendix. Appendix figures and tables should be referred to in the main text as: "Appendix Figure S1, Appendix Figure S2, Appendix Table S1" etc. See detailed instructions regarding expanded view here: <https://www.embopress.org/page/journal/17444292/authorguide#expandedview>.

-Before submitting your revision, primary datasets (and computer code, where appropriate) produced in this study need to be deposited in an appropriate public database (see <http://msb.embopress.org/authorguide-dataavailability> <https://www.embopress.org/page/journal/17444292/authorguide#dataavailability>). Please remember to provide a reviewer password if the datasets are not yet public. The accession numbers and database should be listed in a formal "Data Availability" section (placed after Materials & Method) that follows the model below (see also <https://www.embopress.org/page/journal/17444292/authorguide#dataavailability>). Please note that the Data Availability Section is restricted to new primary data that are part of this study.

Data availability

- RNA-Seq data: Gene Expression Omnibus GSE46843 (<https://www.ncbi.nlm.nih.gov/geo/query/acc.cgi?acc=GSE46843>)

- [data type]: [name of the resource] [accession number/identifier/doi] ([URL or identifiers.org/DATABASE:ACCESSION])

- At EMBO Press we ask authors to provide source data for the main manuscript figures. You will receive a separate email with instructions for providing source data with your revised manuscript, including how to upload and organize the files. Additional information on source data and instruction on how to label the files are available

- Our journal encourages inclusion of *data citations in the reference list* to directly cite datasets that were re-used and obtained from public databases. Data citations in the article text are distinct from normal bibliographical citations and should directly link to the database records from which the data can be accessed. In the main text, data citations are formatted as follows: "Data ref: Smith et al, 2001". In the Reference list, data citations must be labeled with "[DATASET]". A data reference must provide the database name, accession number/identifiers and a resolvable link to the landing page from which the data can be accessed at the end of the reference. Further instructions are available at .

- We updated our journal's competing interests policy in January 2022 and request authors to consider both actual and perceived competing interests. Please review the policy <https://www.embopress.org/competing-interests> and update your competing interests if necessary.

Please use the heading "Disclosure statement and competing interests".

- All Materials and Methods need to be described in the main text using our 'Structured Methods' format. According to this format, the Methods section includes a Reagents and Tools Table (listing key reagents, experimental models, software and relevant equipment and including their sources and relevant identifiers) followed by a Methods and Protocols section describing the methods, ideally using a step-by-step protocol format. The aim is to facilitate adoption of the methodologies across labs. Please download and fill our Reagents and Tools Table template (.docx), which you can find in our author guidelines: <https://www.embopress.org/page/journal/17444292/authorguide#structuredmethods>.

-Regarding data quantification:

Please ensure to specify the name of the statistical test used to generate error bars and P values, the number (n) of independent experiments (please specify technical or biological replicates) underlying each data point and the test used to calculate p-values in each figure legend. Discussion of statistical methodology can be reported in the materials and methods section, but figure legends should contain a basic description of n, P and the test applied.

Graphs must include a description of the bars and the error bars (s.d., s.e.m.).

- Please provide a "standfirst text" summarizing the study in one or two sentences (approximately 250 characters, including space), three to four "bullet points" highlighting the main findings and a "synopsis image" (550px width and 400-600 px height, PNG format) to highlight the paper on our homepage.

Here are a couple of examples:

<https://www.embopress.org/doi/10.15252/msb.20199356>
<https://www.embopress.org/doi/10.15252/msb.20209475>
<https://www.embopress.org/doi/10.15252/msb.209495>

When you resubmit your manuscript, please download our CHECKLIST (<https://www.embopress.org/pb-assets/embo-site/EMBO%20Press%20Author%20Checklist-1642513524327.xlsx>) and include the completed form in your submission. *Please note* that the Author Checklist will be published alongside the paper as part of the transparent process (<https://www.embopress.org/page/journal/17444292/authorguide#transparentprocess>).

If you feel you can satisfactorily deal with these points and those listed by the referees, you may wish to submit a revised version of your manuscript. Please attach a covering letter giving details of the way in which you have handled each of the points raised by the referees. A revised manuscript will be once again subject to review and you probably understand that we can give you no guarantee at this stage that the eventual outcome will be favorable.

Kind regards,
Jingyi

Jingyi Hou, PhD
Senior Editor
Molecular Systems Biology

We realize that it is difficult to revise to a specific deadline. In the interest of protecting the conceptual advance provided by the work, we recommend a revision within 3 months (14th Oct 2025). Please discuss the revision progress ahead of this time with the editor if you require more time to complete the revisions. Use the link below to submit your revision:

*** PLEASE NOTE *** As part of the EMBO Press transparent editorial process initiative (see our Editorial at <https://dx.doi.org/10.1038/msb.2010.72>), Molecular Systems Biology publishes online a Review Process File with each accepted manuscripts. This file will be published in conjunction with your paper and will include the anonymous referee reports, your point-by-point response and all pertinent correspondence relating to the manuscript. If you do NOT want this File to be published, please inform the editorial office at contact@molsystbiol.org within 14 days upon receipt of the present letter.

Reviewer #1:

Unsupervised annotation of cell types in scRNA-seq data is challenging due to the heterogeneity of the data and the lack of a "one-size-fits-all" clustering algorithm. In practice, clustering methods are typically applied without any prior knowledge and clusters are then refined by manually merging or dividing them depending on the expression of certain markers. In this work, the authors propose automating this process by applying a rank-based metric called average overlap (AO) to the ranked lists of DE genes between pairs of clusters. Clusters with high AO scores are merged. The metric is evaluated on T-cell populations and a thymus dataset, and compared against several common alternatives.

Automating the optimization of clustering configurations is of great value to the single-cell community where annotation of cell identities is still very manual and subjective. The AO metric is an attempt to address this gap. While the proposed use of AO for merging and annotating scRNA-seq clusters is clearly described and technically sound, the contribution itself is relatively incremental. In order to establish AO as a superior metric, a significantly more rigorous benchmarking effort is needed by expanding both the set of datasets and the range of baselines used.

Major points

- The proposed AO metric relies heavily on the ranking of top-k DE genes, which makes it sensitive to the DE method used and the choice of k. The current evaluation against prior methods is limited to a single dataset. More complex datasets need to be analyzed to establish the robustness of AO across datasets with batch effects or datasets with more heterogeneous cell type compositions.
- The number of marker genes used for the analysis feels quite small (< 100). Typically, not all DE genes are relevant for

defining cell identity (e.g., housekeeping genes) and their relative ranking may be less meaningful. The authors should discuss the implication of this and perhaps explore more intelligent ways of selecting biologically meaningful DE genes.

- It would be informative to evaluate how well the Leiden algorithm alone is at identifying the correct clusters when varying the resolution parameter. An example of a dataset where Leiden cannot identify the correct clusters at *any* resolution, but a combination of Leiden + AO can, would greatly strengthen the paper. Furthermore, the use of a resolution of 2 for T-cells and 1 for thymus is not fully justified.
- Including a simple metric that computes the direct overlap of the two marker sets (i.e., without ranking) would provide a useful trivial baseline.
- The authors rely on correlation of PCA components for one of their baselines. A more common approach is to perform hierarchical clustering based on distances of these components rather than correlation. It would be useful to see how this baseline compares with the rest.

Minor points

- Several paragraphs in results are primarily methodological and should be moved to the methods section (e.g., lines 133-145, 197-211).
- Figure 1 includes both the method and results. Splitting these would improve readability.

Reviewer #2:

Please refer to the report submitted by my co-referee

Reviewer #3:

The manuscript by C. Thai and colleagues introduces the average overlap (AO) metric as a top-weighted ranked-list similarity measure to quantify similarity between single-cell clusters based on their differentially expressed marker genes. The authors argue that evaluating clusters with AO results in a consistent, precise, and biologically meaningful recapitulation of true cell identities, especially in contexts of subtle transcriptional differences. They benchmark AO on the Zhengmix8eq T-cell dataset with known labels, and then apply it to a mouse thymocyte dataset with unknown cell identities.

Overall, I find the method interesting and potentially well-suited to practical integration into single-cell analysis workflows to aid cluster refinement and merging. However, there are a few issues that should be addressed to clarify its robustness, benchmarking framework, and practical adoption.

Major comments

1. Statistical significance of AO scores

The Methods describe a procedure to assess the statistical significance of AO scores via a null distribution of randomized ranked lists. However, the Results do not report these significance estimates, nor explain how they might impact practical decisions to merge or distinguish clusters. I recommend explicitly reporting these p-values (or adjusted p-values) for the shown AO comparisons, or at least justifying why they were omitted, to help readers interpret the robustness of AO distances.

2. Baseline comparisons

In the Zhengmix8eq benchmark, AO is compared only to other marker-based similarity metrics. To better contextualize its utility, I recommend including two standard baselines commonly used in single-cell workflows: (1) lower-resolution clustering, to assess whether similar groupings could be obtained without a separate refinement step, and (2) dendrograms generated by functions like `scanpy.tl.dendrogram` or Seurat's `BuildClusterTree()`, which operate on global expression or PCA space. These comparisons would clarify whether AO provides unique advantages over default methods already integrated into widely used pipelines.

3. Cell cycle effects and regression

The authors show that AO-based hierarchical clustering in the thymocyte dataset largely recapitulates subpopulations driven by cell cycle phases (e.g., S, G2/M, G1). While this is a biologically interesting finding, it raises the question of whether these differences actually reflect meaningful developmental heterogeneity or are simply proliferative states. It is common in standard scRNA-seq pipelines to regress out cell cycle effects as part of quality control. Therefore, the authors should explicitly justify why they chose to retain cell cycle variation for their AO analysis, or explore whether regressing out these effects would alter the cluster structure. Alternatively, please clarify how confident you are that the developmental annotations derived from AO groupings are not confounded by cell-cycle programs alone.

4. Circularity and benchmarking of AO in the thymocyte analysis (Supplementary Figures 6 and 7)

In the thymocyte dataset, the authors use AO to merge and refine clusters, then describe the resulting annotations as reflecting known thymocyte developmental states. This is valuable, and it is helpful for practitioners to see how AO can guide cluster annotation. However, the subsequent comparison of AO against orthogonal methods, such as pseudotime inference (Supplementary Figure 6) and Pearson correlation-based clustering (Supplementary Figure 7), may be biased. Specifically, these comparisons implicitly test Pearson or pseudotime against cluster identities that were themselves informed by AO, since the marker genes and hierarchical merging structure derive from AO-based groupings. To establish a fair benchmark, I strongly recommend the authors test AO's performance using cluster labels defined independently - for example, through reference-based annotation (e.g., Azimuth, `scmap`) - so that AO, Pearson, and trajectory inference can be compared against the same external standard. This would more convincingly demonstrate that AO recovers true biological relationships beyond those

encoded by its own marker lists.

5. Benchmarking with other correlation metrics

In Supplementary Figure 7, the authors compare AO only to Pearson correlation when evaluating hierarchical clustering of marker gene counts. Since they included Spearman and Kendall-Tau in their earlier benchmarking, it would be helpful for consistency to report these rank-based correlations here as well.

6. Broader benchmarking across heterogeneous datasets

The current validation of AO focuses on two datasets, both composed primarily of T cells, where the main challenge lies in resolving subtle transcriptional differences. While the authors emphasize that AO is designed for such nuanced settings, single-cell RNA-seq datasets in practice vary widely - from closely related populations to highly heterogeneous mixtures such as PBMCs. To support the general applicability of AO, I strongly recommend expanding the benchmarking to include at least one additional dataset involving more distinct cell types (e.g., monocytes, B cells, NK cells). For example, incorporating a formal analysis of the 3k PBMC dataset (currently only included in the GitHub tutorial) within the manuscript would provide valuable insight into how AO performs in well-separated, heterogeneous settings. In any case, demonstrating performance on just two datasets provides limited support for the claimed broad applicability of a new computational method.

Minor comments

1. While the authors demonstrate how AO can be applied after Leiden clustering to merge or refine clusters, they do not explicitly lay out a schematic workflow for readers. Including a short diagram or step-by-step protocol would help clarify how to integrate AO within a typical Scanpy or Seurat pipeline. This would also prevent misinterpretation of AO as a direct cell type annotation method, rather than a cluster refinement tool. For example, some phrasing in the manuscript (e.g. line 95, "methods are lacking for the quantification of marker gene list similarity between single-cell clusters to annotate cell identity") could misleadingly suggest AO itself performs annotation, while in practice it is used to quantify cluster similarities that can inform annotation.

2. In the publicly available PBMC tutorial

(https://github.com/chrisvthai/sc_average_overlap/blob/main/examples/3kpbmcs_tutorial.ipynb), the AO-based hierarchical clustering appears to group cluster 3 with clusters 0 and 1, rather than merging clusters 0-1-4-6 first. This seems somewhat counterintuitive based on the UMAP structure. It would be helpful if the authors could comment on this behavior: is this discrepancy due to distortions inherent to UMAP visualization, or do they believe AO is capturing a biologically meaningful similarity that UMAP does not reflect? If the latter, expanding on this point would help readers better interpret AO results in practice.

Overall, I believe the manuscript addresses a relevant challenge in single-cell data analysis, but would benefit from clarifying how AO fits in practical annotation workflows, ensuring more robust external validation, and transparently reporting the statistical significance of its distance measures.

Point-by-point response to the reviewers' comments:

[AUTHORS] We thank both reviewers for their positive comments and critical evaluation of our work. Here we provide our rebuttal. We hope that both the reviewers and editors will find our revised paper improved and worthy of publication in *Molecular Systems Biology*.

Reviewer #1:

Unsupervised annotation of cell types in scRNA-seq data is challenging due to the heterogeneity of the data and the lack of a "one-size-fits-all" clustering algorithm. In practice, clustering methods are typically applied without any prior knowledge and clusters are then refined by manually merging or dividing them depending on the expression of certain markers. In this work, the authors propose automating this process by applying a rank-based metric called average overlap (AO) to the ranked lists of DE genes between pairs of clusters. Clusters with high AO scores are merged. The metric is evaluated on T-cell populations and a thymus dataset, and compared against several common alternatives.

Automating the optimization of clustering configurations is of great value to the single-cell community where annotation of cell identities is still very manual and subjective. The AO metric is an attempt to address this gap. While the proposed **use of AO for merging and annotating scRNA-seq clusters is clearly described and technically sound**, the contribution itself is relatively incremental. In order to establish AO as a superior metric, a significantly more rigorous benchmarking effort is needed by expanding both the set of datasets and the range of baselines used.

[AUTHORS] We thank the reviewer for her/his overall positive evaluation and constructive criticism of our work.

Major points

- The proposed AO metric relies heavily on the ranking of top-k DE genes, which makes it sensitive to the DE method used and the choice of k. The current evaluation against prior methods is limited to a single dataset. More complex datasets need to be analyzed to establish the robustness of AO across datasets with batch effects or datasets with more heterogeneous cell type compositions.

[AUTHORS] In response to this reviewer's concern, we have expanded our benchmarking to include more datasets (in addition to the Zheng subset) that meet our criteria of using ground truths that are defined independently of the scRNA-seq assay itself. Originally, we chose the subset of T-cells in the Zheng dataset because of its complex and challenging

nature, with ground truth labels that were determined independently with a separate assay (cell sorting). We now utilize both the complete Zhengmix8eq dataset that includes more distinct populations, such as monocytes and B-cells, and its subset of T-cells that we used originally. We also include a CITE-seq dataset offering both cell surface protein and transcriptome information, with cell identities derived from optimized clustering based on protein data. We reasoned that, in a similar manner to the Zheng dataset, transcriptional similarity between clusters should align well with the similarity of clusters using protein data. Indeed, our results show the consistency of AO-based analysis compared to all tested metrics for inferring cell populations that match ground truth cell identities derived orthogonal and independent of scRNA-seq measurements. Detailed results on newly added datasets are now included in the manuscript (**lines 148-157**) and in the figures (**New Fig 2; New Fig EV1**).

Additionally, we have introduced a new functional measure of clustering that aims to take the gene expression profiles of each inferred cluster label into account. This was implemented as another more general measure of gene expression similarity between inferred clusters and ground truth populations that was applicable to each of our new datasets used. Using this revised benchmarking workflow that builds off on what was introduced in the original manuscript, we noticed that in each dataset there is at least one hierarchical clustering scheme that performs similarly to using average overlap. However, the comparative performance of these other clustering schemes, like using Pearson correlation on either marker gene counts or principal components, is a singular occurrence in a single dataset. Meanwhile, the average overlap metric consistently performs high across all datasets used for benchmarking, all while providing a simple interpretation of cluster marker gene list similarity (**New Fig 2; New Fig EV2**).

We did not focus on the effect of batch effects on our benchmarking because data integration of multiple experiments is outside the scope of our post-cluster analysis framework. Batch effect correction and data integration normally occurs upstream of cluster analysis. If no such measures are performed, then cells in such datasets would likely cluster based on batch, and any measure of cluster similarity, not just average overlap, will take into account gene expression differences corresponding to batch.

- The number of marker genes used for the analysis feels quite small (< 100). Typically, not all DE genes are relevant for defining cell identity (e.g., **housekeeping genes**) and their relative ranking may be less meaningful. The authors should discuss the implication of this and perhaps explore more intelligent ways of selecting biologically meaningful DE genes.

[AUTHORS] In response to this reviewer's comment, we have expanded the number of marker genes in our analyses in all the datasets that we use for benchmarking. Our results

now show that the application of average overlap with 200 or 500 marker genes does not yield a significantly different performance compared to using only up to 100 marker genes (**New Figure EV3**). Thus, we argue that using less than 100 marker genes per cluster is sufficient to explain cluster separation.

It is true that the exact expression of housekeeping genes is not by itself relevant for establishing cell identity. However, in the context of scRNA-seq clustering, if a set of what are normally considered housekeeping genes is significantly differentially expressed in one cluster versus the rest, then this measured difference in gene expression might be a driver of cell identity. As a result, such genes in that particular dataset would not be stably expressed across all cells any longer, which is the entire premise of housekeeping genes. As these differential expression tests are performed in a cluster-versus-rest manner to derive marker genes, any significantly DE gene constitutes important biological variation. Thus, we consider it is best not filter genes out of our analysis on the basis of whether they are largely considered housekeeping genes.

- It would be informative to evaluate how well the Leiden algorithm alone is at identifying the correct clusters when varying the resolution parameter. An example of a dataset where Leiden cannot identify the correct clusters at *any* resolution, but a combination of Leiden + AO can, would greatly strengthen the paper. Furthermore, the use of a resolution of 2 for T-cells and 1 for thymus is not fully justified.

[AUTHORS] As suggested by all reviewers, we have now included using lower-resolution clustering alone without any subsequent cluster refinement step as the baseline for which all other clustering configurations are benchmarked against (**New Fig 2; New Fig EV1-2; New Appendix Fig S7**). For this baseline, in each dataset used, we chose clustering resolutions that give the same number of clusters as there are ground truth populations. It should be noted that the assignment of cells to clusters even at fixed resolutions is nondeterministic in nature and some cells may be assigned to different clusters when Leiden clustering is rerun. In contrast, and as shown in **New Fig. 2**, average overlap enables consistent and accurate assignment of cells in truth-known data independent of Leiden clustering initial seed or resolutions. Moreover, we have expanded our analysis to include clustering the cells using the Leiden algorithm at various resolutions (1.0–2.0 for *Zhengmix8eq*, 1.5–2.0 for the T-cell subset, and 0.6–1.2 for the CBMCs) (**New Fig EV2; New Appendix Fig S5-6**). In all benchmarked datasets, we now show that AO + Leiden clustering produces consistent results independent of clustering resolution without the need for fine-tuning Leiden to achieve the exact number of ground truth populations.

- Including a simple metric that computes the direct overlap of the two marker sets (i.e., without ranking) would provide a useful trivial baseline.

[AUTHORS] In response to this important comment, and as noted above, we have now expanded benchmarking and show direct overlap between marker gene sets as a baseline in **New Appendix Fig S7**. In the case of highly dissimilar clusters a common gene is not expected until increasing number of genes are used to define the marker gene sets. Therefore, our results showed that utilizing direct overlap of cluster marker gene sets, performed extremely poorly across all datasets when <50 number of marker genes were included. In contrast, when >100 marker genes were included, direct overlap and AO showed similar performance in capturing true cell identities in all datasets.

- The authors rely on correlation of PCA components for one of their baselines. A more common approach is to perform hierarchical clustering based on distances of these components rather than correlation. It would be useful to see how this baseline compares with the rest.

[AUTHORS] In response to this reviewer's valid concern, we have now added the Euclidean distance as a metric for comparing PCs (**New Fig 2, New Fig EV2-3, New Appendix Fig S2-6**), which performs similarly to correlation-based metrics, and inferior to AO-based analyses.

Minor points

- Several paragraphs in results are primarily methodological and should be moved to the methods section (e.g., lines 133-145, 197-211).

- Figure 1 includes both the method and results. Splitting these would improve readability.

[AUTHORS] We agree with the reviewer. Thus, we have trimmed and moved text from the results section that was more appropriate for the methods sections (new lines 484, 498, 501-506). Additionally, we have now split up the previous Figure 1 into what is currently **New Figure 1**, describing average overlap and our proposed workflow to incorporate it into clustering analysis, and **New Figure 2**, which details more of the benchmarking results we generated with new datasets.

Reviewer #2:

Please refer to the report submitted by my co-referee

Reviewer #3:

The manuscript by C. Thai and colleagues introduces the average overlap (AO) metric as a top-weighted ranked-list similarity measure to quantify similarity between single-cell clusters based on their differentially expressed marker genes. The authors argue that evaluating clusters with AO results in a consistent, precise, and biologically meaningful recapitulation of true cell identities, especially in contexts of subtle transcriptional differences. They benchmark AO on the Zhengmix8eq T-cell dataset with known labels, and then apply it to a mouse thymocyte dataset with unknown cell identities.

Overall, I find the method interesting and potentially well-suited to practical integration into single-cell analysis workflows to aid cluster refinement and merging. However, there are a few issues that should be addressed to clarify its robustness, benchmarking framework, and practical adoption.

[AUTHORS] We thank the reviewer for her/his overall positive evaluation and constructive criticism of our work.

Major comments

1. Statistical significance of AO scores

The Methods describe a procedure to assess the statistical significance of AO scores via a null distribution of randomized ranked lists. However, the Results do not report these significance estimates, nor explain how they might impact practical decisions to merge or distinguish clusters. I recommend explicitly reporting these p-values (or adjusted p-values) for the shown AO comparisons, or at least justifying why they were omitted, to help readers interpret the robustness of AO distances.

[AUTHORS] As rightfully pointed out by the reviewer, we previously did not emphasize the significance estimates of average overlap distances and how these may impact cluster comparison in practice. We now provide a distribution of z-scores of the pairwise AO scores of ground truth clusters in one of our benchmarking datasets (**New Fig EV4**), to provide an example of how to interpret significance of AO scores in real biological populations (**New lines 134-138, 204-213**). Additionally, we report AO z-scores for AO distances across all our results for the benchmarking and thymocyte analyses (**New Appendix Fig S8**).

2. Baseline comparisons

In the Zhengmix8eq benchmark, AO is compared only to other marker-based similarity metrics. To better contextualize its utility, I recommend including two standard baselines

commonly used in single-cell workflows: (1) lower-resolution clustering, to assess whether similar groupings could be obtained without a separate refinement step, and (2) dendrograms generated by functions like `scanpy.tl.dendrogram` or Seurat's `BuildClusterTree()`, which operate on global expression or PCA space. These comparisons would clarify whether AO provides unique advantages over default methods already integrated into widely used pipelines.

[AUTHORS] As suggested by all reviewers, we have now included using lower-resolution clustering alone, without any subsequent cluster refinement step, as the baseline for which all other clustering configurations are benchmarked against.

We currently already use dendrograms generated by `scanpy.tl.dendrogram()` in our benchmarking process, or we have implemented cluster trees that closely follow the implementation given by the scanpy developers. The dendrograms generated using scanpy by default enable the use of Pearson, Kendall, and Spearman correlations, and utilize either expression counts of a user-specified set of genes or PCA space. These combinations are what we already used for benchmarking AO (**New Fig 2; New Fig EV2; New Appendix Fig S2-6**).

Seurat's `BuildClusterTree()` function similarly builds a tree that is estimated on a distance matrix constructed in either gene expression space or PCA space with the option to specify either. We have additionally benchmarked the use of the Euclidean distance in constructing dendrograms, which closely follows Seurat's description of their `BuildClusterTree()` function (**New Fig 2; New Fig EV2; New Appendix Fig S2-6**). Overall, our results show that the AO metric performed very similarly to the baseline obtained from a fixed, highly optimized Leiden clustering that produced the same number of clusters as ground truth populations. Across all datasets and initial Leiden resolutions tested, the AO metric applied to cluster marker genes had the best performance compared to other metrics, whether calculated on expression counts or principal components.

3. Cell cycle effects and regression

The authors show that AO-based hierarchical clustering in the thymocyte dataset largely recapitulates subpopulations driven by cell cycle phases (e.g., S, G2/M, G1). While this is a biologically interesting finding, it raises the question of whether these differences actually reflect meaningful developmental heterogeneity or are simply proliferative states. It is common in standard scRNA-seq pipelines to regress out cell cycle effects as part of quality control. Therefore, the authors should explicitly justify why they chose to retain cell cycle variation for their AO analysis, or explore whether regressing out these effects would alter the cluster structure. Alternatively, please clarify how confident you are that the developmental annotations derived from AO groupings are not confounded by cell-cycle programs alone.

[AUTHORS] As the reviewer pointed out, it is a common practice to regress out cell cycle effects as part of quality control. This holds true when cell cycle variation is unwanted in the dataset –in an atlas-type dataset with many heterogeneous, well-separated cell populations, one would not want multiple of the same cell types, only differing in cell cycle phase. However, the context of thymocyte development makes the cell cycle a potentially important factor to consider when analyzing gene expression variation. T-cells pass different development checkpoints, especially during the double negative stages, and proliferate extensively after passing such checkpoints.

In response to this reviewer's comment, we now also performed cell cycle regression as defined in scanpy by using the *sc.tl.regress_out* function with the cell cycle scores we had already calculated. We expectedly lost separation within many of the double negative stages of development that we had characterized previously (**New Figure EV5**). This can be seen when we overlay both our original annotations as well as annotations provided by using the SingleR tool with labels of sorted thymocytes from ImmGen RNA-seq datasets.

We believe that the proliferative states that we find in our data are in fact important identifying features of developing thymocytes, and directly inform cluster separation when using the Leiden algorithm. As a result, we argue the cell cycle signature should not be regressed out of the data, especially in this context. We also argue that the fact that cell cycle has historically been regressed out might have been part of the reason why identifying subpopulations of DN thymocytes using scRNAseq was notoriously difficult/impossible. We have further emphasized this in the Discussion section (**New lines 396-412**).

4. Circularity and benchmarking of AO in the thymocyte analysis (Supplementary Figures 6 and 7)

In the thymocyte dataset, the authors use AO to merge and refine clusters, then describe the resulting annotations as reflecting known thymocyte developmental states. This is valuable, and it is helpful for practitioners to see how AO can guide cluster annotation.

However, the subsequent comparison of AO against orthogonal methods, such as pseudotime inference (Supplementary Figure 6) and Pearson correlation-based clustering (Supplementary Figure 7), may be biased. Specifically, these comparisons implicitly test Pearson or pseudotime against cluster identities that were themselves informed by AO, since the marker genes and hierarchical merging structure derive from AO-based groupings. To establish a fair benchmark, I strongly recommend the authors test AO's performance using cluster labels defined independently - for example, through reference-based annotation (e.g., Azimuth, scmap) - so that AO, Pearson, and trajectory inference can be compared against the same external standard. This would more

convincingly demonstrate that AO recovers true biological relationships beyond those encoded by its own marker lists.

[AUTHORS] We apologize for not making this point clearer in our first submission. In our original paper, our brief orthogonal analysis with pseudotime and generating cluster trees using Pearson correlation of marker gene expression counts was conducted with the original Leiden cluster labels in mind. Our additional cluster trees using Spearman, Kendall-tau, and Euclidean distances are also generated using the original Leiden labels (**New Appendix Fig S13**). These orthogonal analyses are not downstream of any cluster refinement involving AO.

We already use the RNA expression profile of sorted bulk populations of thymocytes in the ImmGen project as our reference for assigning cluster identities after refinement with AO-based hierarchical clustering. Still, as an additional point of comparison, we annotated individual cells with the SingleR tool, using the very same reference dataset of sorted bulk RNA-seq data of thymocytes in ImmGen (**New Appendix Fig S14**). This automated annotation identifies stages of double negative thymocyte development, but not to the resolution that we achieve with a more unsupervised approach incorporating cluster refinement with AO. Clusters in our data (as we explain in our previous response to major comment #3) are identified heavily by their cell cycle signatures, which show up in their most differentially expressed genes.

5. Benchmarking with other correlation metrics

In Supplementary Figure 7, the authors compare AO only to Pearson correlation when evaluating hierarchical clustering of marker gene counts. Since they included Spearman and Kendall-Tau in their earlier benchmarking, it would be helpful for consistency to report these rank-based correlations here as well.

[AUTHORS] We agree with the reviewer. In response, we have now added cluster trees generated with Spearman and Kendall-Tau correlations of clusters' marker gene counts as implemented with *scanpy's sc.tl.dendrogram* function, as well as the tree generated with Euclidean distance in terms of marker gene counts using our implementation closely following Seurat's *BuildClusterTree* function (**New Appendix Fig S13**).

6. Broader benchmarking across heterogeneous datasets

The current validation of AO focuses on two datasets, both composed primarily of T cells, where the main challenge lies in resolving subtle transcriptional differences. While the authors emphasize that AO is designed for such nuanced settings, single-cell RNA-seq datasets in practice vary widely - from closely related populations to highly heterogeneous mixtures such as PBMCs. To support the general applicability of AO, I strongly

recommend expanding the benchmarking to include at least one additional dataset involving more distinct cell types (e.g., monocytes, B cells, NK cells). For example, incorporating a formal analysis of the 3k PBMC dataset (currently only included in the GitHub tutorial) within the manuscript would provide valuable insight into how AO performs in well-separated, heterogeneous settings. In any case, demonstrating performance on just two datasets provides limited support for the claimed broad applicability of a new computational method.

[AUTHORS] The reviewer raises an interesting point. We originally refrained from using the widely-used 3k PBMC dataset in our benchmarking as there is no experimentally-derived ground truth. While the 3k PBMC data has been characterized, clustered, and annotated repeatedly and robustly, we aimed to focus our benchmarks on datasets where ground-truth labels could be established with a simultaneous measurement with an assay different from RNA-seq, such as through cell sorting with methods like FACS or CITE-seq. In general, datasets where labels are commonly assigned to cell populations through clustering and expert annotation of marker genes could potentially show bias in our benchmarking toward distance metrics or methods that were incorporated in the original clustering analysis used to obtain the ground truth to begin with.

To answer this specific concern, we have benchmarked AO and alternative distance metrics on the complete Zhengmix8eq dataset, which contains sorted monocytes, B-cells, and natural killer cells, in addition to the T-cell subset we had originally chosen. Additionally, we also utilized a CITE-seq dataset of cord blood mononuclear cells, where we fixed a ground-truth labeling of the cells using clustering of the protein measurements (**New Fig EV1**). In both cases, we maintain our criteria for using ground-truth labels derived from an assay separate from cells' RNA measurements. Our choice of using the entire Zhengmix8eq dataset as well as the refined CBMC CITE-seq dataset should provide insight into how AO performs with well-separated, heterogeneous settings, and not just high-similar cell-types that we showed previously with our T-cell subsets during benchmarking, and thymocyte development in our application of the AO method.

Minor comments

1. While the authors demonstrate how AO can be applied after Leiden clustering to merge or refine clusters, they do not explicitly lay out a schematic workflow for readers. Including a short diagram or step-by-step protocol would help clarify how to integrate AO within a typical Scanpy or Seurat pipeline. This would also prevent misinterpretation of AO as a direct cell type annotation method, rather than a cluster refinement tool. For example, some phrasing in the manuscript (e.g. line 95, "methods are lacking for the quantification of marker gene list similarity between single-cell clusters to annotate cell identity") could

misleadingly suggest AO itself performs annotation, while in practice it is used to quantify cluster similarities that can inform annotation.

[AUTHORS] We previously provided a schematic workflow of how to incorporate AO and hierarchical clustering into a typical scRNA-seq analysis in old Figure 1b. We have since split up our first figure panel into two parts, now with **New Figure 1** being solely to explain the AO metric and our proposed workflow for incorporating it into clustering analysis for scRNA-seq data. Meanwhile, **New Figure 2** displays more of the benchmarking results we have generated, including the new datasets that we have added.

We have clarified phrasing in the manuscript to ensure AO is described as a tool for cluster comparison, and not for direct cluster annotation itself (such as in old line 95, and elsewhere).

2. In the publicly available PBMC tutorial

(https://github.com/chrisvthai/sc_average_overlap/blob/main/examples/3kpbmcs_tutorial.ipynb), the AO-based hierarchical clustering appears to group cluster 3 with clusters 0 and 1, rather than merging clusters 0-1-4-6 first. This seems somewhat counterintuitive based on the UMAP structure. It would be helpful if the authors could comment on this behavior: is this discrepancy due to distortions inherent to UMAP visualization, or do they believe AO is capturing a biologically meaningful similarity that UMAP does not reflect? If the latter, expanding on this point would help readers better interpret AO results in practice.

[AUTHORS] We thank the reviewer for their careful review and this interesting observation. We acknowledge that there is a mismatch between the visual representation and grouping of clusters in the cluster tree and their actual similarity as defined by the AO metric. Indeed, looking at the pairwise distance heatmap, the AO z-scores between clusters 0 and 3 or 1 and 3 are very similar to those between 0,1 and 4,6. As described above (under “1. Statistical significance of AO scores”), we have clarified the use of the AO z-scores on the secondary y-axis of cluster trees. In this specific example with PBMCs, when unifying cluster marker genes before ranking, we see that while cluster 3 is visually grouped with the pair of clusters 0 and 1, this branch corresponds to a z-score of near 0, suggesting that there is poor significance of AO similarity. Instead, the single pairs of clusters 4 with 6 and clusters 0 and 1 should be grouped first, as their AO scores correspond to z-scores of greater than 10. Afterwards, the remaining pairs of clusters (i.e., the next branch of the cluster tree) have comparatively lower AO distances, as signified by their z-scores. As the reviewer points out, UMAP projection may put cells in clusters 0, 1, 4, and 6 in closer spatial proximity compared to those in cluster 3, but transcriptionally, these cells are in fact distinct from each other.

Overall, I believe the manuscript addresses a relevant challenge in single-cell data analysis, but would benefit from clarifying how AO fits in practical annotation workflows, ensuring more robust external validation, and transparently reporting the statistical significance of its distance measures.

[AUTHORS] We thank again the reviewer for her/his comments, which have overall helped us strengthen the relevance of our workflow and findings.

19th Nov 2025

Manuscript Number: MSB-2025-13193R

Title: CANTAO: GUIDING CLUSTERING AND ANNOTATION IN SINGLE-CELL RNA SEQUENCING USING AVERAGE OVERLAP

Author: Christopher Thai

Amartya Singh

Daniel Herranz

Hossein Khiabani

Dear Dr Herranz,

Thank you for submitting the revised version of your manuscript to Molecular Systems Biology. We have now received the enclosed reports from the reviewers who agreed to re-assess it. As you will see, the original Reviewer #1 (now Reviewer #3) is satisfied with the revisions. The original Reviewer #3 (now Reviewer #2) also finds the manuscript substantially improved; however, they think the concerns regarding the interpretation of the cell-cycle effects remain insufficiently addressed.

Therefore, before we can formally accept the paper for publication, we kindly ask that you provide a response to this remaining comment from Reviewer #2, and we recommend tempering the claims that "cell-cycle phase is a distinguishing feature for stages of thymocyte development" or "strongest driver" by using a more cautious phrasing.

On a more editorial level, please do the following:

1. Please remove all figures from the manuscript file and upload them as individual, high-resolution figure files. Figure legends should be placed below the References.
2. Include the "Funding" information within the "Acknowledgements" section and remove the separate "Funding" heading.
3. Rename "CONFLICTS OF INTEREST" to "DISCLOSURE AND COMPETING INTERESTS STATEMENT."
4. Provide up to five keywords in the manuscript file.
5. Remove the "Author Contribution" section from the manuscript file.
6. Add missing callouts for Figure 3E and individual panels for EV and Appendix figures.
7. Data availability:
 - Only primary data generated in this study should be declared; references to previously published datasets should be removed from this section.
 - Remove the "Code availability" heading and include the relevant information under "Data availability."
8. Address the following issues in figure legends:
 - Please note that the box plots need to be defined in terms of minima, maxima, centre, bounds of box and whiskers, and percentile in the legends of figures 2A-F; EV2 A-C; EV3 A-C
 - Please note that information related to n is missing in the legends of figures 2A-F; EV2 A-C; EV3 A-C
9. Sections need to be named and the order should be corrected: Title page - Abstract - Keywords - Introduction - Results - Discussion - Methods - Data Availability - Acknowledgements - Disclosure and Competing Interests Statement - References - Figure Legends - Table(s) - Expanded View Figure Legends.

Click on the link below to submit your revised paper.

Sincerely,

Jingyi

Jingyi Hou, PhD
Senior Editor
Molecular Systems Biology

*** PLEASE NOTE *** As part of the EMBO Press transparent editorial process initiative (see our Editorial at <https://dx.doi.org/10.1038/msb.2010.72> , Molecular Systems Biology will publish online a Review Process File to accompany accepted manuscripts. When preparing your letter of response, please be aware that in the event of acceptance, your cover letter/point-by-point document will be included as part of this File, which will be available to the scientific community. More information about this initiative is available in our Instructions to Authors. If you have any questions about this initiative, please contact the editorial office (msb@embo.org).

Reviewer #1:

Please refer to the report submitted by my co-referee

Reviewer #2:

The revised manuscript by Thai et al. presents the average overlap (AO) metric for quantifying similarity between single-cell clusters based on ranked marker genes - now in the form of the CANTAO method (Clustering and ANnotation using Transcriptomic Average Overlap). We find the revisions responsive and the manuscript substantially improved; our previous concerns have been addressed with an important exception regarding the interpretation of cell cycle effects: the conclusion that "cell-cycle phase is a distinguishing feature for stages of thymocyte development" appears overstated. The observed merging of DN2a/b and DN4/ISP clusters after regressing out cell-cycle-associated genes more likely reflects that the original clustering was dominated by proliferation-related transcriptional variation, rather than indicating that cell-cycle phase itself defines developmental identity. DN2a-DN2b and DN3b-DN4/ISP transitions are characterized not only by proliferative bursts but also by molecular changes (eg lineage commitment, β -selection, pre-TCR activation <https://pmc.ncbi.nlm.nih.gov/articles/PMC3994230/>). These transcriptional and signaling features (e.g. *Il2ra*, *Ptcra*, *Cd28*) should distinguish the stages independently of cell-cycle status. Therefore, we believe this particular analysis remains incomplete.

Reviewer #3:

I thank the authors for their response. This revised version of the manuscript is improved with the addition of new datasets and new experiments, in particular, the new baselines with varying Leiden resolution and varying choices of overlap k. I have no further comments.

Point-by-point response to the reviewers' comments:

[AUTHORS] We thank both reviewers for their positive comments and critical evaluation of our work. Here we provide our response to the last outstanding comment.

Reviewer #1:

Please refer to the report submitted by my co-referee

Reviewer #2:

The revised manuscript by Thai et al. presents the average overlap (AO) metric for quantifying similarity between single-cell clusters based on ranked marker genes - now in the form of the CANTAO method (Clustering and ANnotation using Transcriptomic Average Overlap). **We find the revisions responsive and the manuscript substantially improved;**

[AUTHORS] We would like to thank the reviewer for her/his positive evaluation of our revision.

our previous concerns have been addressed with an important exception regarding the interpretation of cell cycle effects: the conclusion that "cell-cycle phase is a distinguishing feature for stages of thymocyte development" appears overstated. The observed merging of DN2a/b and DN4/ISP clusters after regressing out cell-cycle-associated genes more likely reflects that the original clustering was dominated by proliferation-related transcriptional variation, rather than indicating that cell-cycle phase itself defines developmental identity. DN2a-DN2b and DN3b-DN4/ISP transitions are characterized not only by proliferative bursts but also by molecular changes (eg lineage commitment, β -selection, pre-TCR activation <https://pmc.ncbi.nlm.nih.gov/articles/PMC3994230/>). These transcriptional and signaling features (e.g. Il2ra, Ptcr, Cd28) should distinguish the stages independently of cell-cycle status. Therefore, we believe this particular analysis remains incomplete.

[AUTHORS] To complete this part of our analysis, we have now carefully analyzed the expression of the key factors in these transitions, and it is important to note that canonical protein markers of T-cell developmental stages are generally not differentially expressed, transcriptionally speaking (**Figure for reviewers 1**). Indeed, Log-normalized gene expression counts for a select set of thymocyte markers show minimal variation between DN2a, 2b, 3a, and 3b populations (**Figure for reviewers 1A**). When looking at the top 1000 differentially expressed genes of our final annotated populations, ranked by

A

B

	Bcl11b	Il2ra	Kit	Cd44	Cd27	Cd28	Ptcr
MPP/DN1				320			
DN2a		14	42				
DN2b		190	742				
DN3a	255	2	778				23
DN3b		147	821				
DN4/ISP (S phase)							
DN4/ISP (G2/M phase)							
DN4/ISP (G2/M phase) - A							
ISP							
ISP to DP					446		
DP	46				55	332	
CD4 T					49	7	
CD8 T				524		761	
NK cell				63		816	
gd T	502				338		

adjusted p-values for the Wilcoxon ranksum test, most of them are undetectable or fall outside of the top 100 rankings for most of these difficult-to-call populations, with the exception of Il2ra (i.e. cd25) and Kit in DN2a and/or DN3a cells (**Figure for reviewers 1B**). This indicates that, while these canonical markers at the protein level are certainly enough to assign T-cell developmental stages, the relative and ranking of differential expression of genes outside of this set might be more informative when trying to assign T-cell developmental stage from transcriptional data alone, especially when using scRNAseq data. Overall, removing the cell cycle effects abrogates our

capability to accurately differentiate DN2a cells from DN2b cells, while the DN3b/DN4/ISP populations are still separated but in a much less confident way than when including the cell cycle effects (**Fig. 3E** and **Fig. EV5E**).

Still, we agree that our previous conclusion was overstated, and we have therefore adjusted the main text and discussion accordingly (**see revised lines 265-268, 353-355, 403-406 and 426-432**). We would like to once again thank this reviewer for her/his careful assessment of our work, which has significantly helped us to improve our manuscript.

Reviewer #3:

I thank the authors for their response. This revised version of the manuscript is improved with the addition of new datasets and new experiments, in particular, the new baselines with varying Leiden resolution and varying choices of overlap k. I have no further comments.

[AUTHORS] We thank again the reviewer for her/his positive assessment and prior comments, which overall helped us strengthen the relevance of our workflow and findings.

24th Nov 2025

Manuscript number: MSB-2025-13193RR

Title: CANTAO: GUIDING CLUSTERING AND ANNOTATION IN SINGLE-CELL RNA SEQUENCING USING AVERAGE OVERLAP

Dear Dr Herranz,

Thank you again for sending us your revised manuscript. We are now satisfied with the modifications made and I am pleased to inform you that your paper has been accepted for publication.

Sincerely,
Jingyi

Jingyi Hou, PhD
Senior Editor
Molecular Systems Biology
